# Computationally-driven identification of antibody epitopes

**Casey K Hua[1,2], Albert T Gacerez[2], Charles L Sentman[2], Margaret E Ackerman[1,2], Yoonjoo Choi[3]\*, Chris Bailey-Kellogg[4]\***

[1]Thayer School of Engineering, Dartmouth College, Hanover, United States; [2]Department of Microbiology and Immunology, Geisel School of Medicine, Dartmouth College, Lebanon, United States; [3]Department of Biological Sciences, Korea Advanced Institute of Science and Technology (KAIST), Daejeon, Republic of Korea; [4]Department of Computer Science, Dartmouth College, Hanover, United States

**Abstract** Understanding where antibodies recognize antigens can help define mechanisms of action and provide insights into progression of immune responses. We investigate the extent to which information about binding specificity implicitly encoded in amino acid sequence can be leveraged to identify antibody epitopes. In computationally-driven epitope localization, possible antibody–antigen binding modes are modeled, and targeted panels of antigen variants are designed to experimentally test these hypotheses. Prospective application of this approach to two antibodies enabled epitope localization using five or fewer variants per antibody, or alternatively, a six-variant panel for both simultaneously. Retrospective analysis of a variety of antibodies and antigens demonstrated an almost 90% success rate with an average of three antigen variants, further supporting the observation that the combination of computational modeling and protein design can reveal key determinants of antibody–antigen binding and enable efficient studies of collections of antibodies identified from polyclonal samples or engineered libraries.
DOI: https://doi.org/10.7554/eLife.29023.001

**\*For correspondence:**
yoonjoo.choi@kaist.ac.kr (YC);
cbk@cs.dartmouth.edu (CB-K)

## Introduction

Antibodies have long been recognized for their beneficial roles in vaccination, infection, and clinical therapy, as well as their pathogenic roles in autoimmunity. The protective and/or pathogenic capacity of an antibody (Ab) is functionally delimited by the specific epitope(s) that it recognizes on an antigen (Ag). Thus, even Abs targeting the same Ag have demonstrated variable efficacy dependent upon their epitope specificities. In cancer, Abs against particular epitopes have demonstrated increased therapeutic effects and decreased off-tumor toxicities (*Garrett et al., 2009*; *Gan et al., 2012*; *Kim et al., 2002*), and combinations of mAbs targeting diverse epitopes have demonstrated synergistic action and delayed the development of treatment resistance (*Koefoed et al., 2011*; *Friedman et al., 2005*). Similarly, Abs against particular epitopes have been associated with protection in the setting of vaccination (*Zolla-Pazner et al., 2014*; *Gottardo et al., 2013*; *Steel et al., 2010*; *Gocník et al., 2007*; *Liu et al., 2016*; *Eggink et al., 2014*; *Margine et al., 2013*) and natural infection (*Walker et al., 2010*; *Lu et al., 2016*). As a result, the identification of epitopes contributing to potent antibody bioactivity is rapidly gaining attention in vaccine design efforts (*Haynes, 2015*; *West et al., 2014*; *Zolla-Pazner et al., 2016*; *Correia et al., 2014*; *He et al., 2015*; *Lanzavecchia et al., 2016*; *Pica and Palese, 2013*; *Subbarao and Matsuoka, 2013*) as well as in reducing immunogenic responses against protein drug candidates (*Nagata and Pastan, 2009*; *Onda et al., 2011*; *Onda et al., 2008*).

While characterization of epitope specificities is important for both scientific investigation and clinical translation, the epitopes targeted by newly isolated Abs are often unknown. In the therapeutic setting, novel Ag-specific antibodies are typically discovered through in vitro selections or in vivo immunizations using whole Ag proteins (*Lu et al., 2012*; *Nelson et al., 2010*). Such efforts generate multiple Ab candidates simultaneously, which may target multiple different (and unknown) regions on the Ag. Since only a limited number of candidates may be taken forward, it may be helpful at this early stage to distinguish their modes of recognition, which in turn may impact their mechanisms of action (*Ditto and Brooks, 2016*; *Brooks et al., 2014*). Similarly, in the vaccine setting, Abs purified from subject sera may target a wide range of epitopes, some previously determined but some with potent new modes of binding, potentially conferring different protective mechanisms (*Zolla-Pazner, 2004*; *Lewis, 2010*; *Khurana et al., 2016*). Because next generation sequencing and more robust Ab discovery platforms are greatly expanding the repertoire of Ag-specific Abs with known sequences (*Doria-Rose et al., 2014*; *Liao et al., 2013*; *Wu et al., 2015*), efficient identification of epitopes from Ab sequence is a highly attractive target (*Robinson, 2015*) by which to monitor the immune response, characterize the development of Ab repertoires, and potentially lead to novel discoveries and new therapies.

To characterize Ab epitopes, structure determination (*Saul and Alzari, 1996*) is the gold standard (*Abbott et al., 2014*), but may be impractical in early stages or in investigations involving multiple Abs. Consequently, a variety of other methods have been developed that trade off resolution in favor of reduced time and expense. For example, epitope-binning assays (*Ditto and Brooks, 2016*; *Brooks et al., 2014*) can compare tens or even hundreds of Abs at a time, but are currently limited in resolution to identifying only which specificities overlap. Site-directed mutagenesis approaches, such as alanine scanning (*Weiss et al., 2000*), have become relatively routine (*Greenspan and Di Cera, 1999*) and can identify specific Ag residues critical for Ab binding, but require expression and testing of fairly large numbers of variants. Recently, Kowalsky et al. scaled this approach significantly, demonstrating that a comprehensive combination of mutagenesis, surface display, and deep sequencing can provide fast and effective fine epitope mapping (*Kowalsky et al., 2015*). Spectrometry-based methods such as HDX-MS (*Gallagher and Hudgens, 2016*; *Huang and Chen, 2014*) and NMR (*Zuiderweg, 2002*) similarly offer very detailed resolution of residues involved, but require expensive equipment and specific expertise in processing and analyzing results and may be subject to protein size limitations.

In comparison to experimental efforts, computational analysis is efficient and inexpensive, and thus much effort has gone into developing methods to predict epitopes in silico. Many methods have attempted to perform epitope prediction in the absence of any information about particular antibodies (reviewed in [*Gao and Kurgan, 2014*; *Zhang et al., 2014*]), in order to help predict the overall immunogenicity of an Ag. Unfortunately, recent collaborative efforts between computational and experimental researchers have suggested that since any Ag surface region has the potential to serve as an epitope, such predictions may be of limited practical utility (*Sela-Culang et al., 2015*). In contrast, computational efforts to predict epitopes for specified Abs (given the amino acid sequence and potentially a structure or homology model) address this concern and have recently made substantial progress (*Sela-Culang et al., 2015*; *Zhao and Li, 2010*; *Zhao et al., 2011*; *Soga et al., 2010*; *Brenke et al., 2012*; *Krawczyk et al., 2014*; *Sircar and Gray, 2010*; *Sela-Culang et al., 2013*). While promising, purely computational methods are not yet sufficiently reliable to stand alone (*Sela-Culang et al., 2013*; *Yao et al., 2013*). Thus, a recently proposed paradigm integrates computational and experimental methods, leveraging the advantages of each (*Sela-Culang et al., 2015*). Integrated methods to date are predicated on the availability of initial experimental information to improve computational predictions, which are subsequently experimentally tested to definitively identify epitopes (*Kowalsky et al., 2015*; *Araya and Fowler, 2011*; *Chuang et al., 2013*; *Sela-Culang et al., 2014*).

We investigate the extent to which the Ab-Ag recognition information encoded in the proteins themselves can be harnessed to drive the entire epitope localization process. In particular, we study the ability of computational methods to optimize experimental validation of computational predictions, thereby focusing and minimizing experimental effort. Ab-Ag docking benchmarks have shown that at least one near-native docking model can usually be found among generated samples; unfortunately, such methods fail to reliably indicate which one (*Brenke et al., 2012*). However, the models can be viewed as hypotheses (though not necessarily mutually exclusive) to be experimentally tested

via site-directed mutagenesis and binding assays. Through EpiScope, an integrated computational-experimental approach (*Figure 1*), Ag variants are designed for each docking model such that, if a model is consistent with the true binding mode, Ab binding will be ablated in the corresponding Ag variant(s). The variants are distilled to a small set representing all docking models such that if any of the models is correct, one of the variants will fail to bind the Ab. Experimental identification of a variant with disrupted binding enables localization of the epitope to include one or more of the mutated residues. The docking models are then filtered to a smaller set (there need not be a unique one) representing binding modes and corresponding footprints on the Ag that are consistent with the effects of the disruptive mutations. As we demonstrate in prospective application to two Abs against a tumor antigen, as well as thorough retrospective testing with a wide range of Ab-Ag pairs, Ab sequence and Ag structure alone are sufficient to drive efficient targeting of experimental effort to effectively localize epitopes. We further demonstrate that decoding binding information from sequence enables the multiplexing of experimental epitope localization efforts for multiple Abs targeting the same Ag.

## Results

### Computationally-driven Ab epitope identification: EpiScope

The integrated computational-experimental framework is described in *Figure 1* (full details are provided as a PyMol session file, *Supplementary file 1*) and was implemented as follows. Ab–Ag

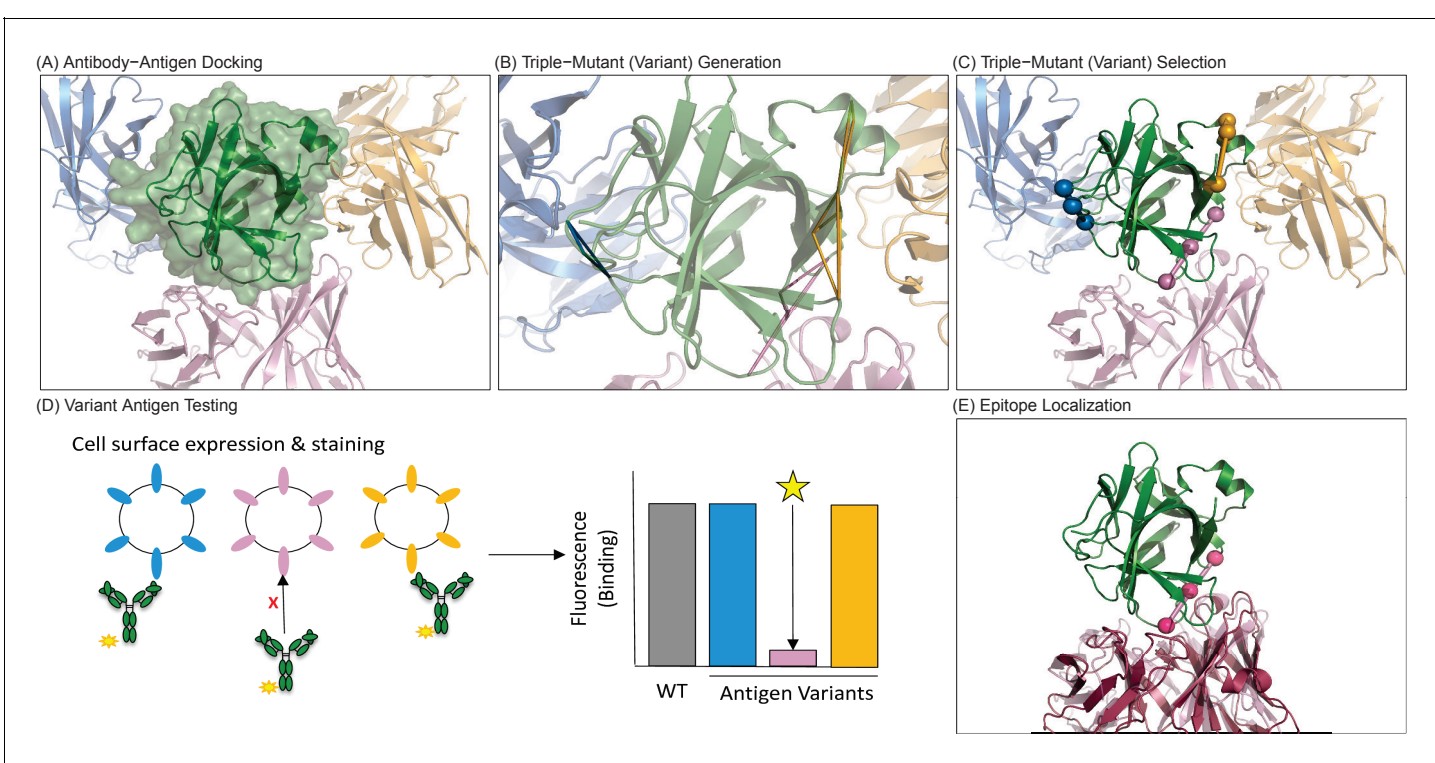

**Figure 1.** Overview of computationally-driven epitope identification by EpiScope. (A) Ab–Ag docking models are generated using computational docking methods. In the example, the green structure is the Ag human IL-18 (PDB ID: 2VXT:A), while the cartoons represent possible poses of the Ab (limited here to three for clarity). Full details including docking models and designs for this example are provided in a PyMol session (*Supplementary file 1*). (B) Ag variants containing a pre-defined numbers of mutations (here triple mutations, colored triangles) are generated for each docking model. (C) Variants are clustered with respect to spatial locations in the Ag, and a set of variants predicted to disrupt all of the docking models is selected. (D) Ag mutagenesis and Ag-Ab binding experiments are performed to identify which mutations result in loss of Ab recognition. (E) Examination of the disruptive variant(s) enables localization of the Ab epitope in terms of both mutated positions (pink balls) and consistent docking models, here with the model (light pink cartoon) quite similar to the actual crystal structure (dark pink cartoon).
DOI: https://doi.org/10.7554/eLife.29023.002

docking models were generated by the ClusPro server (*Brenke et al., 2012*; *Comeau et al., 2004a*), which in a recent Ab–Ag docking benchmark (*Brenke et al., 2012*) demonstrated a near-native docking model among its top 30 predictions in 95% of test cases. For each model, site-directed mutagenesis based Ag variants were computationally designed (*Choi et al., 2017*; *Parker et al., 2013*) to disrupt Ab binding, as evaluated by a sequence potential (*Pons et al., 2011*), while maintaining Ag stability, as evaluated by molecular mechanics modeling (*Gainza et al., 2013*; *Pearlman et al., 1995*). These two properties were balanced in a Pareto optimal fashion (*He et al., 2015*), with the goal of ensuring that variants still express and fold similarly to the wild type protein, thereby enabling confident interpretation of Ab binding results. Designed variants were clustered to identify a minimal set predicted to disrupt binding to all of the docking models, ensuring coverage of all computational hypotheses. The selected Ag designs were experimentally evaluated for retention or loss of Ab binding, where loss of Ab binding signal suggests overlap of the true Ab epitope with at least one of the designed mutations in that variant.

## Epitope localization of two monoclonal Abs

Our investigation of computationally-driven epitope mapping was prompted by studies of two previously uncharacterized antibodies against tumor Ag B7H6: TZ47, a murine antibody generated through mouse immunizations (*Choi et al., 2015*), and PB11, a human scFv generated through directed evolution of a human Ab fragment library (*Feldhaus et al., 2003*). EpiScope designed a set of 4 (for TZ47) and 5 (for PB11) triple-mutant B7H6 variants to probe all of the docking models (*Figure 2A and D* and *Table 1*. Design details including docking models are provided in a PyMol session file in *Supplementary file 2*, and all sequence details are provided in *Supplementary file 3*). There were 28 docking models for each Ab, and each variant was predicted to disrupt between 5 and 12 of the models. The designed B7H6 variants were expressed in the context of the full-length transmembrane protein on the surface of human embryonic kidney (HEK) cells and evaluated for Ab binding via flow cytometry. All variants maintained binding to NKp30, the natural ligand for B7H6 (*Figure 2B and E*), suggesting proper expression and folding. Lower NKp30 binding signal for PB-Ag2 may result from the proximity of designed mutations to the NKp30 binding site, rather than changes in Ag expression or stability. Conversely, a lack of binding to negative control antibodies with alternative Ag specificities demonstrated that the introduced mutations did not facilitate indiscriminate Ab binding. Altogether, these findings suggest that binding changes for designed variants resulted from specific disruption of Ab binding interfaces, rather than altered Ag stability or structure.

Two Ab-specific designs, TZ47-Ag4 and PB11-Ag2, reduced Ab binding to levels comparable to the negative control while maintaining binding to the natural ligand (*Figure 2B and E*), suggesting that at least one of the three designed mutations in each variant is part of the epitope. While this constitutes successful epitope localization, the results can be further interpreted in terms of the docking models with binding interfaces disrupted by these mutations. There were five such models for each Ab (out of the original 28 each), substantially limiting the possible epitope regions to 17.4% and 26.7% of the surface respectively (out of the original 82% and 88% covered by the sets of docking models) (*Figure 2C and F*). Thus, a small set of designs localized the epitope on the Ag in terms of disruptive mutations and the footprints of the docking models consistent with those effects.

In some cases, it may be desirable to pursue follow-up experiments to obtain finer resolution of the epitope guided by the initial coarse-grained localization. Here, a chimera-based approach was used to further probe the TZ47 epitope, based on the identified disruptive design TZ47-Ag4 along with prior experimental results demonstrating that TZ47 cannot recognize macaque B7H6 Ag (despite ~75% identity to human B7H6). The chimera SD9 (*Figure 2—figure supplement 1*) contains the macaque B7H6 sequence in the region of the designed mutations in TZ47-Ag4, differing from the human sequence by four amino acids including TZ47-Ag4's mutation at M154, where the macaque sequence contained a similarly hydrophobic valine (V) and TZ47-Ag4 contained a more dramatic change to a negatively charged glutamate (E). Chimera SD9 similarly disrupted TZ47 binding, reconfirming the importance of the common mutation site and general epitope region.

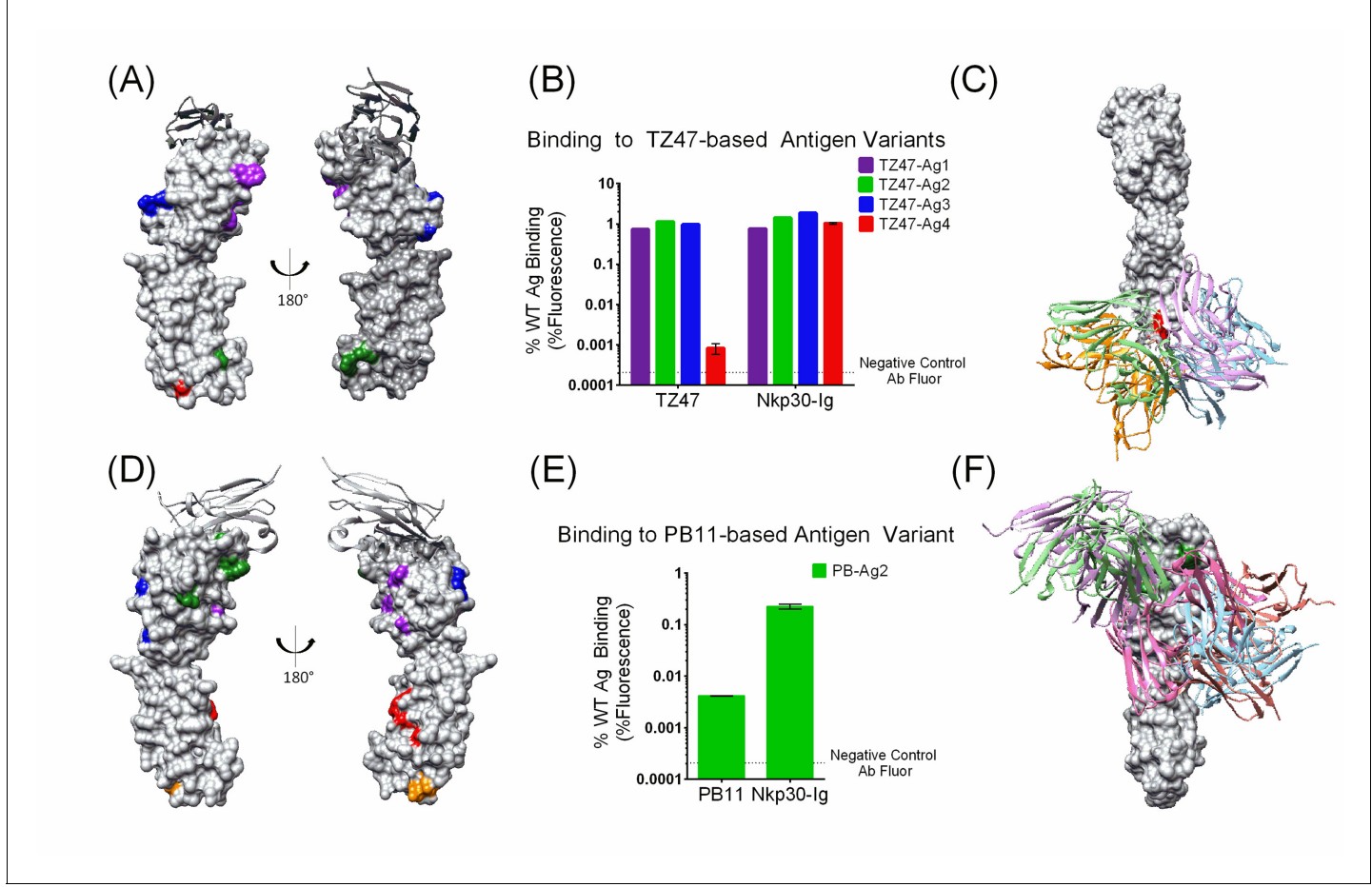

**Figure 2.** Small sets of designed Ag variants enable epitope localization for two different B7H6-targeting Abs. (A–C). TZ47; (D-F) PB11. (A and D) Designed Ag variants, color-coded by triple mutation sets (*Table 1*). NKp30, a natural ligand for B7H6, is shown in grey ribbon. (B and E) Flow cytometry results from staining variant-expressing HEK cells with the relevant Ab, using NKp30-Ig as a positive control. Fluorescence is normalized to WT Ag-expressing cells. The dotted lines represent average background fluorescence measured from negative control Abs. Experiments were conducted in triplicate and error bars show the standard deviation. (C and F) Docking models (Ab cartoons of different colors) affected by the disruptive Ag variants (highlighted in red for TZ47 and green for PB11). Bar graphs depict the average (height) and standard deviation (error bars) of the MFI of 3 technical replicates, defined as the equivalent staining of a single batch of transfected cells repeated in three separate wells in the same experiment. One outlier value was excluded (PB11-staining of PB11-Ag1) where fewer than 1500 live cells were sampled and the raw MFI was two orders of magnitude higher than the other two replicates (1145.6 vs. 14.41 and 14.20).

DOI: https://doi.org/10.7554/eLife.29023.003

The following source data and figure supplement are available for figure 2:

**Source data 1.** Raw data values for *Figure 2B and E*.
DOI: https://doi.org/10.7554/eLife.29023.005
**Figure supplement 1.** Chimeric variant (SD9) design confirmed the localization of the TZ47 epitope.
DOI: https://doi.org/10.7554/eLife.29023.004

## Ag variant design for simultaneous localization of both Abs

Despite the large distance separating the localized epitopes of TZ47 and PB11 (*Figure 3—figure supplement 1*), significant overlap was observed between the initial docking models for the two Abs (*Figure 3—figure supplement 2*). This overlap led to similarities in the Ag variant designs, with TZ47-targeted designs covering 23/28 PB11 docks and PB11-targeted designs covering 25/28 TZ47 docks. These results suggested that the Ab-specificity of docking models is limited and that greater experimental efficiency could be achieved by optimizing designs to disrupt predicted epitopes common to the Abs.

**Table 1.** Summary of mutations in EpiScope Ag designs for each Ab.
Designs that disrupted binding for each Ab are highlighted.

| Design | Mutations |
|---|---|
| TZ47-Ag1 | F47Y, N49Q, W98E |
| TZ47-Ag2 | F184D, I188Q, V225T |
| TZ47-Ag3 | T71K, K74E, V76H |
| TZ47-Ag4 | M154E, N157G, S217H |
| PB-Ag1 | M30V, Q132V, Q136L |
| PB-Ag2 | F51H, Y52D, R99G |
| PB-Ag3 | A88T, F89T, G111R |
| PB-Ag4 | T176K, V194I, R231E |
| PB-Ag5 | N216K, S217A, Q219V |

DOI: https://doi.org/10.7554/eLife.29023.007

To determine if experiments could be designed to take advantage of similarities in possible binding modes while also accounting for differences, we generated an integrated set of 6 designs (*Table 2*) interrogating all 56 docking models generated for the two antibodies combined (*Figure 3A*; full details are provided in a PyMol session in *Supplementary file 2*). This simultaneous design scheme represents a substantial reduction from the initial total of 9 designs (four for TZ47 and 5 for PB11) to separately localize each epitope, but still resulted in successful localization of both Abs, with two different variants successfully disrupting binding to the two Abs (*Figure 3B*). These disruptive variants overlap five docking models each (*Figure 3C*), the majority of which were also affected by disruptive designs based on individual Abs, demonstrating agreement in the localization of TZ47 and PB11 epitopes from both single Ab-input and multiple Ab-input designs to a few docking models. We conclude that the multi-Ab approach can both decrease the required experimental effort and increase the likelihood of successful epitope localization, offering the novel capacity to multiplex epitope localization efforts through the rational design of Ag variant panels to simultaneously probe multiple Ab inputs.

## Generalizability of localizing Ab epitopes from sequence-encoded information

To assess the generalizability of harnessing Ab sequence-encoded binding information to design efficient epitope localization experiments, we designed Ag variant sets for 33 distinct Ab-Ag complexes with high quality crystal structures (*Table 3*). In these tests, an Ab epitope was considered successfully localized if at least one of the generated designs contained a mutation within the Ab-Ag binding interface. By this metric, the epitope was successfully localized in 88% (29/33) of the test cases (*Figure 4A*). Strikingly, this success rate could be achieved using an average of only 3 Ag variants for each test case (*Figure 4B*). There was a weak correlation between Ag size and the number

**Table 2.** Summary of mutations in Multi-Ab specific EpiScope Ag designs.
Designs that disrupted binding for each Ab are highlighted.

| Design | Mutations |
|---|---|
| MULTI-1 | N57D, D84N, W98E (PB11) |
| MULTI-2 | F66Y, T71K, F72D |
| MULTI-3 | V78L, F89T, G111R |
| MULTI-4 | M154E, N157E, N216K (TZ47) |
| MULTI-5 | A172H, R231E, A233E |
| MULTI-6 | T176K, R231E, H236S |

DOI: https://doi.org/10.7554/eLife.29023.012

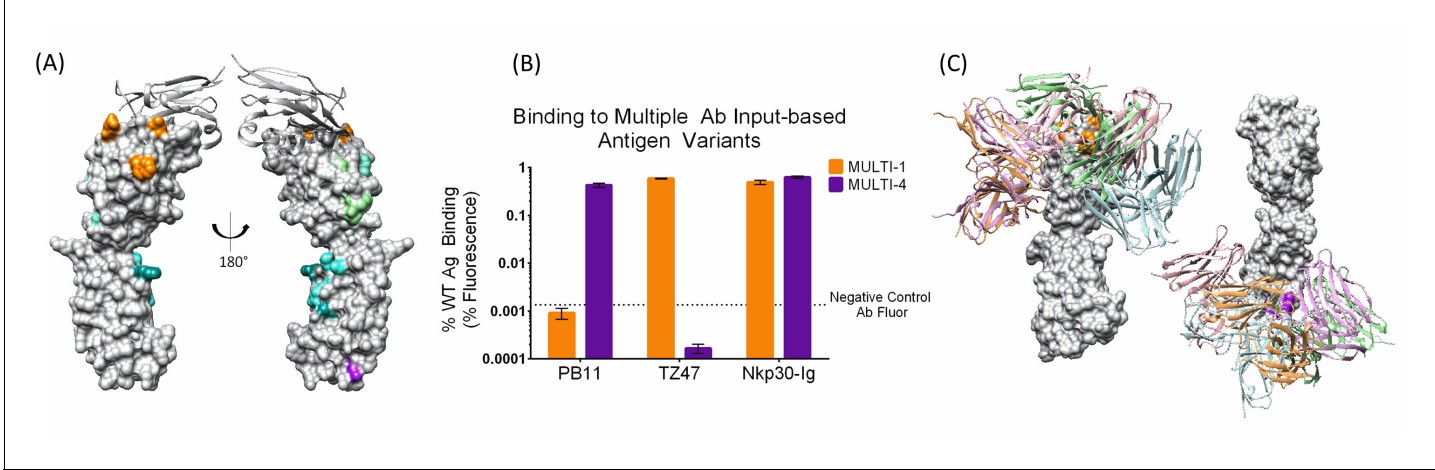

**Figure 3.** A single set of Ag variants enables simultaneous localization of two different B7H6-targeting Abs. (**A**) Designed Ag variants color-coded by triple-mutant design, with natural binding partner NKp30 in grey ribbon. (**B**) Flow cytometry results from staining variant-expressing HEK cells with the relevant Ab, using NKp30 as a positive control. Fluorescence was normalized to WT antigen-expressing cells. The dotted line represents average background fluorescence measured from negative control Abs. (**C**) Docking models (Ab cartoons of different colors) affected by disruptive Ag variants (highlighted in orange for TZ47 and magenta for PB11), for left: TZ47 and right: PB11. Bar graphs depict the average (height) and standard deviation (error bars) of the MFI of 3 technical replicates, defined as the equivalent staining of a single batch of transfected cells repeated in three separate wells in the same experiment. One replicate value was excluded where fewer than 1500 live cells were sampled from the well (one replicate of PB11-staining of MULTI-1) and the raw MFI was two orders of magnitude larger than the other two replicates (232.8 vs. 2.55 and 3.71).
DOI: https://doi.org/10.7554/eLife.29023.008

The following source data and figure supplements are available for figure 3:

**Source data 1.** Raw data values for *Figure 3B*.
DOI: https://doi.org/10.7554/eLife.29023.011
**Figure supplement 1.** B7H6 and epitope localization for its binding antibodies.
DOI: https://doi.org/10.7554/eLife.29023.009
**Figure supplement 2.** Docking models for TZ47 and PB11 substantially overlap.
DOI: https://doi.org/10.7554/eLife.29023.010

of experiments needed to probe all generated docking models (r = 0.51), but no correlation between Ag size and the rate of successful epitope localization (*Figure 4—figure supplement 1*).

We proceeded to use this dataset to investigate the impacts of the various inputs and parameters on the effectiveness of epitope localization.

## Ag homology models

We investigated whether the information contained in Ag sequence would suffice to drive epitope localization via homology modeling, or whether the Ag crystal structure was required. For extra stringency, Ag homology models were based on moderately similar template structures (20–50% sequence identity, which yielded models with relatively high structural similarity, average TM scores (*Zhang and Skolnick, 2004*) of 0.82). Ag homology models were equally effective to crystal structures, obtaining an 85% success rate (vs. 88% for crystal structures) in epitope localization (*Figure 4A* and *Figure 4—figure supplement 1*), and still requiring an average of only three variants (*Figure 4B*). Thus a homology model can provide a suitable surrogate for Ag structure when no crystal structure is available. Surprisingly, use of the homology model enabled localization of two Ab epitopes missed when using the crystal structure, but failed to localize three Ab epitopes captured by use of the crystal structure. In these cases, the most native-like docking model generated for the failed Ag structure was less similar to the true binding mode (as measured by a lower $f_{nat}$ score (*Lensink et al., 2007*)) than that generated using the alternative Ag structure (*Tables 3*, *4* and *5*).

## Random designs

In order to evaluate how much the process benefits from docking models, we performed the same design approach but using random sets of surface positions (still constraining average Cα

**Table 3.** Retrospective test cases.

Columns indicate the PDB ID of each Ab-Ag pair; the number of residues for various subsets of the Ag; the number and success of EpiScope designs based on crystal and model Ag structures; a measure of the quality of the closest native-like docking model among ClusPro generated models ($f_{nat}$[*Lensink et al., 2007*]); the quality of the homology models built for Ab and Ags (TM-score [*Zhang and Skolnick, 2004*]); and the number of docking decoys generated by ClusPro.

| | Number of residues | | | Crystal structure | | Model structure | | Fnat | | TM-score | | Number of docking decoys | |
|---|---|---|---|---|---|---|---|---|---|---|---|---|---|
| PDB code | Whole | Surface | Epitopes | Number of designs | Overlap with epitopes | Number of designs | Overlap with epitopes | Crystal | Model | Antibody | Antigen | Crystal | Model |
| 1FE8 | 196 | 124 | 27 | 4 | Y | 3 | N | 0.1 | 0.04 | 0.96 | 0.84 | 30 | 24 |
| 1FNS | 196 | 120 | 12 | 5 | Y | 2 | Y | 0.39 | 0.09 | 0.96 | 0.86 | 26 | 20 |
| 1H0D | 123 | 96 | 14 | 3 | N | 2 | Y | 0 | 0.05 | 0.98 | 0.79 | 30 | 30 |
| 1LK3 | 160 | 102 | 26 | 3 | Y | 3 | Y | 0.73 | 0.44 | 0.97 | 0.74 | 23 | 29 |
| 1OAZ | 123 | 101 | 14 | 2 | N | 2 | N | 0.1 | 0.1 | 0.97 | 0.77 | 30 | 29 |
| 1OB1 | 99 | 74 | 13 | 3 | Y | 4 | Y | 0.62 | 0.61 | 0.97 | 0.85 | 30 | 29 |
| 1RJL | 95 | 82 | 13 | 3 | Y | 2 | Y | 0.29 | 0.3 | 0.96 | 0.89 | 30 | 27 |
| 1V7M | 163 | 113 | 20 | 6 | Y | 3 | Y | 0.45 | 0.19 | 0.96 | 0.78 | 27 | 24 |
| 1YJD | 140 | 86 | 14 | 2 | Y | 3 | Y | 0.42 | 0.13 | 0.98 | 0.77 | 25 | 30 |
| 2ARJ | 123 | 90 | 17 | 3 | Y | 3 | Y | 0.63 | 0.26 | 0.97 | 0.75 | 24 | 30 |
| 2VXQ | 96 | 71 | 21 | 3 | Y | 3 | Y | 0.32 | 0.21 | 0.89 | 0.89 | 30 | 30 |
| 2VXT | 157 | 116 | 19 | 3 | Y | 3 | Y | 0.83 | 0.13 | 0.95 | 0.93 | 13 | 19 |
| 2XQB | 114 | 87 | 18 | 2 | Y | 3 | Y | 0.16 | 0.25 | 0.92 | 0.89 | 17 | 21 |
| 3D9A | 129 | 93 | 19 | 3 | Y | 2 | Y | 0.09 | 0.63 | 0.93 | 0.93 | 22 | 29 |
| 3HI1 | 290 | 246 | 20 | 4 | N | 3 | N | 0.28 | 0.02 | 0.97 | 0.83 | 30 | 29 |
| 3L5X | 113 | 83 | 8 | 6 | Y | 3 | Y | 0.18 | 0.24 | 0.98 | 0.83 | 30 | 30 |
| 3MXW | 169 | 108 | 22 | 2 | Y | 2 | Y | 0.52 | 0.47 | 0.96 | 0.92 | 20 | 23 |
| 3QWO | 57 | 48 | 10 | 3 | Y | 4 | Y | 0.32 | 0.53 | 0.96 | 0.86 | 30 | 30 |
| 3RKD | 146 | 105 | 18 | 5 | Y | 3 | Y | 0.55 | 0.48 | 0.97 | 0.49 | 30 | 30 |
| 4DN4 | 76 | 50 | 12 | 2 | Y | 2 | Y | 0.49 | 0.28 | 0.92 | 0.88 | 20 | 27 |
| 4DW2 | 222 | 175 | 20 | 4 | Y | 3 | Y | 0.1 | 0.35 | 0.92 | 0.85 | 30 | 30 |
| 4ETQ | 226 | 186 | 22 | 4 | Y | 3 | Y | 0.45 | 0.57 | 0.96 | 0.94 | 30 | 30 |
| 4G3Y | 157 | 114 | 12 | 3 | Y | 4 | Y | 0.35 | 0.12 | 0.94 | 0.86 | 30 | 30 |
| 4G6J | 158 | 109 | 13 | 3 | Y | 3 | Y | 0.57 | 0.15 | 0.97 | 0.85 | 30 | 30 |
| 4I3S | 190 | 163 | 23 | 4 | Y | 4 | Y | 0.05 | 0.09 | 0.81 | 0.82 | 30 | 30 |
| 4JZJ | 252 | 210 | 18 | 4 | Y | 6 | Y | 0.31 | 0.25 | 0.95 | 0.33 | 30 | 30 |
| 4KI5 | 183 | 108 | 7 | 1 | Y | 1 | Y | 0.39 | 0.29 | 0.95 | 0.93 | 24 | 20 |
| 4L5F | 111 | 79 | 9 | 2 | Y | 3 | Y | 0.52 | 0.1 | 0.97 | 0.8 | 30 | 30 |
| 4LVH | 223 | 184 | 13 | 5 | Y | 6 | N | 0.12 | 0.05 | 0.93 | 0.9 | 30 | 29 |
| 4M62 | 155 | 105 | 6 | 2 | Y | 2 | N | 0.12 | 0.11 | 0.92 | 0.81 | 24 | 24 |
| 4NP4 | 272 | 230 | 25 | 3 | Y | 5 | Y | 0.09 | 0.07 | 0.96 | 0.67 | 30 | 30 |
| 4RGO | 226 | 187 | 17 | 3 | N | 6 | Y | 0.16 | 0.21 | 0.97 | 0.96 | 30 | 30 |
| 5D96 | 235 | 198 | 22 | 4 | Y | 4 | Y | 0.23 | 0.15 | 0.95 | 0.96 | 30 | 30 |
| Average | 162.88 | 122.52 | 16.48 | 3.30 | | 3.18 | | 0.33 | 0.24 | 0.95 | 0.82 | 27.12 | 27.67 |
| STD | 57.57 | 51.53 | 5.54 | 1.19 | | 1.21 | | 0.21 | 0.18 | 0.03 | 0.13 | 4.53 | 3.55 |

DOI: https://doi.org/10.7554/eLife.29023.021

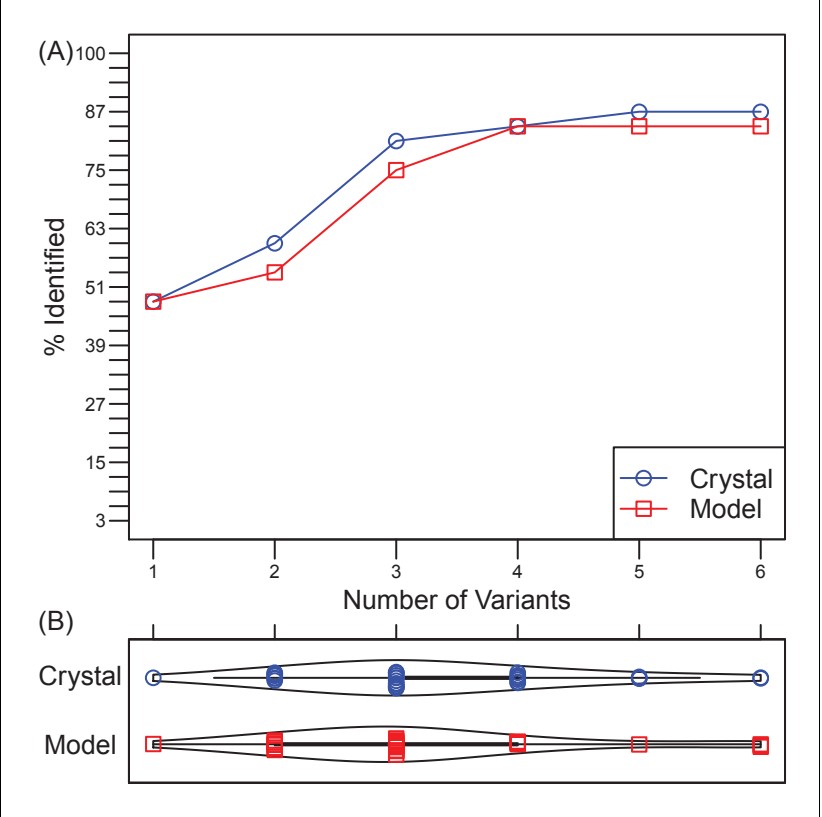

**Figure 4.** Retrospective validation demonstrates generality of efficiency and effectiveness in localizing epitopes. (A) Over a test set of 33 diverse Ab-Ag pairs with co-crystal structures, the number of pairs in which at least one binding interface residue is included among the disruptive mutations in a set of 1–6 Ag triple-mutant variants. Ultimately, two pairs were missed when using Ag crystal structure and three pairs when using Ag homology models. (B) Violin plots of the number of Ag variants required to incorporate mutations predicted to disrupt all docking models.

DOI: https://doi.org/10.7554/eLife.29023.013

The following source data and figure supplements are available for figure 4:

**Figure supplement 1.** Ag size vs. number of Ag variants to cover docking models.
DOI: https://doi.org/10.7554/eLife.29023.014

**Figure supplement 2.** Performance using size-matched sets of random triple-mutants instead of docking-based disruptive ones.
DOI: https://doi.org/10.7554/eLife.29023.015

**Figure supplement 2—source data 1.** Raw data values for *Figure 4—figure supplement 2*.
DOI: https://doi.org/10.7554/eLife.29023.018

**Figure supplement 3.** Varying the number of mutations to include per design from 1 to 4 demonstrates that the most efficient epitope localization occurs at three mutations/design.
DOI: https://doi.org/10.7554/eLife.29023.016

**Figure supplement 3—source data 2.** Raw data values for *Figure 4—figure supplement 3*.
DOI: https://doi.org/10.7554/eLife.29023.019

**Figure supplement 4.** Effects of inter-mutation distances on epitope localization resolution and success rate.
DOI: https://doi.org/10.7554/eLife.29023.017

**Figure supplement 4—source data 3.** Raw data values for *Figure 4—figure supplement 4*.
DOI: https://doi.org/10.7554/eLife.29023.020

distance <12 Å) instead of using designs optimized to disrupt specific docking models. Here surface was defined as a relative solvent accessibility greater than 7% (*Mizuguchi et al., 1998*). For each target, 1000 random triple mutants were generated and subsets were selected by the same clustering approach so as to match the number of plans used by EpiScope for that target. To account for

**Table 4.** Ab modeling quality.

Antibody structures were generally highly accurately predicted both overall (average TM-score: 0.95) and for CDRs (all-backbone-atom, including N, C, Cα and O, RMSDs reported). Overall, non-CDR-H3 loops were very well predicted based on the canonical rules, and even for CDR-H3 loops the average RMSDs was <2 Å.

| Target | Species | CDR-L1 | L2 | L3 | H1 | H2 | CDR-H3 RMSD | Sequence | Length | TM-score |
|--------|---------|--------|----|----|----|----|------|----------|--------|----------|
| 1FE8 | MOUSE | 0.42 | 0.22 | 0.74 | 1.01 | 0.51 | 0.63 | AGNYYGMDY | 9 | 0.96 |
| 1FNS | MOUSE | 0.54 | 0.18 | 0.93 | 0.27 | 0.60 | 2.10 | VRDPADYGNYDYALDY | 16 | 0.96 |
| 1H0D | MOUSE | 1.43 | 0.57 | 0.42 | 0.44 | 1.11 | 0.66 | TRLGDYGYAYTMDY | 14 | 0.98 |
| 1LK3 | RAT | 0.41 | 0.43 | 0.52 | 0.57 | 1.51 | 1.00 | TRGVPGNNWFPY | 12 | 0.97 |
| 1OAZ | MOUSE | 1.15 | 0.44 | 0.88 | 1.30 | 0.56 | 1.25 | ARMWYYGTYYFDY | 13 | 0.97 |
| 1OB1 | MOUSE | 0.58 | 0.31 | 0.63 | 0.42 | 0.63 | 1.97 | ARNYYRFDGGMDF | 13 | 0.97 |
| 1RJL | MOUSE | 1.43 | 0.57 | 4.96 | 0.69 | 1.00 | 1.16 | ARMRYGDYYAMDN | 13 | 0.96 |
| 1V7M | MOUSE | 0.70 | 0.26 | 0.83 | 0.65 | 1.10 | 0.59 | SGWSFLY | 7 | 0.96 |
| 1YJD | MOUSE | 0.88 | 0.51 | 1.34 | 0.62 | 1.19 | 1.76 | TRSHYGLDWNFDV | 13 | 0.98 |
| 2ARJ | RAT | 0.71 | 0.67 | 1.12 | 0.46 | 0.70 | 0.65 | TPLIGSWYFDF | 11 | 0.97 |
| 2VXQ | HUMAN | 0.35 | 0.74 | 0.96 | 0.90 | 1.27 | 1.05 | ARLDGYTLDI | 10 | 0.89 |
| 2VXT | MOUSE | 0.47 | 0.37 | 1.14 | 0.45 | 0.53 | 0.43 | ARGLRF | 6 | 0.95 |
| 2XQB | HUMAN | 1.61 | 0.43 | 0.98 | 1.19 | 0.89 | 7.21 | ARDPAAWPLQQSLAWFDP | 18 | 0.92 |
| 3D9A | MOUSE | 0.40 | 0.61 | 1.18 | 0.99 | 1.88 | 0.51 | ANWDGDY | 7 | 0.93 |
| 3HI1 | HUMAN | 0.80 | 0.86 | 0.83 | 0.61 | 0.44 | 1.25 | ARGPVPAVFYGDYRLDP | 17 | 0.97 |
| 3L5X | HUMAN | 0.56 | 0.61 | 0.91 | 1.05 | 0.90 | 1.73 | ARMGSDYDVWFDY | 13 | 0.98 |
| 3MXW | HUMAN | 0.58 | 0.71 | 0.71 | 1.09 | 0.82 | 0.96 | ARDWERGDFFDY | 12 | 0.96 |
| 3QWO | HUMANIZED | 0.48 | 0.28 | 1.09 | 0.87 | 0.50 | 1.13 | ARDMIFNFYFDV | 12 | 0.96 |
| 3RKD | MOUSE | 0.62 | 0.42 | 0.52 | 1.06 | 0.65 | 1.45 | ARIKSVITTGDYALDY | 16 | 0.97 |
| 4DN4 | HUMAN | 2.11 | 0.37 | 1.58 | 1.64 | 2.40 | 2.36 | ARYDGIYGELDF | 12 | 0.92 |
| 4DW2 | MOUSE | 1.20 | 0.43 | 4.12 | 0.85 | 1.14 | 3.18 | ERGELTYAMDY | 11 | 0.92 |
| 4ETQ | MOUSE | 1.07 | 0.29 | 1.59 | 0.35 | 0.91 | 0.94 | TRSNYRYDYFDV | 12 | 0.96 |
| 4G3Y | CHIMERIC | 0.68 | 0.71 | 0.57 | 0.90 | 0.98 | 1.22 | SRNYYGSTYDY | 11 | 0.94 |
| 4G6J | HUMAN | 0.72 | 0.44 | 0.90 | 0.41 | 0.35 | 1.14 | ARDLRTGPFDY | 11 | 0.97 |
| 4I3S | HUMAN | 1.34 | 0.46 | 0.64 | 4.33 | 1.08 | 3.49 | ARQKFYTGGQGWYFDL | 16 | 0.81 |
| 4JZJ | HUMAN | 0.59 | 0.54 | 0.98 | 0.84 | 1.04 | 2.96 | ARSHLLRASWFAY | 13 | 0.95 |
| 4KI5 | MOUSE | 0.74 | 0.52 | 0.78 | 2.14 | 0.44 | 1.49 | AREDDGLAS | 9 | 0.95 |
| 4L5F | MOUSE | 0.76 | 0.42 | 1.03 | 0.49 | 0.94 | 1.83 | TKRINWALDY | 10 | 0.97 |
| 4LVH | MOUSE | 1.63 | 0.71 | 2.82 | 1.46 | 2.70 | 1.91 | ARHGSPGYTLYAWDY | 15 | 0.93 |
| 4M62 | HUMAN | 2.08 | 0.79 | 1.40 | 2.55 | 2.78 | 8.26 | AREGTTGSGWLGKPIGAFAY | 20 | 0.92 |
| 4NP4 | HUMAN | 2.21 | 0.87 | 2.84 | 0.88 | 0.55 | 1.53 | ARRRNWGNAFDI | 12 | 0.96 |
| 4RGO | MOUSE | 0.53 | 1.01 | 0.75 | 0.71 | 0.31 | 2.20 | VRDLYGDYVGRYAY | 14 | 0.97 |
| 5D96 | MOUSE | 0.74 | 0.57 | 0.53 | 0.62 | 0.89 | 3.43 | ASDSMDPGSFAY | 12 | 0.95 |
| Average | | 0.92 | 0.52 | 1.25 | 0.99 | 1.01 | 1.92 | | | 0.95 |
| STD | | 0.53 | 0.20 | 1.01 | 0.78 | 0.62 | 1.71 | | | 0.03 |

DOI: https://doi.org/10.7554/eLife.29023.022

The following source data available for  Table 4:

Source data 1.
DOI: https://doi.org/10.7554/eLife.29023.023

**Table 5.** The quality of Ag models and their template structures.
Failed cases are highlighted in red.

| Target | Template | Template chain | Seq. ID. | TM-score |
|--------|----------|----------------|----------|----------|
| 1FE8 | 3PPY | A | 28.09 | 0.84 |
| 1FNS | 4IGI | A | 24.73 | 0.86 |
| 1H0D | 3MWQ | A | 33.88 | 0.79 |
| 1LK3 | 4DOH | A | 27.94 | 0.74 |
| 1OAZ | 2PUK | C | 48.04 | 0.77 |
| 1OB1 | 1N1I | A | 49.44 | 0.85 |
| 1RJL | 2FKJ | C | 62.11 | 0.89 |
| 1V7M | 1CN4 | C | 23.74 | 0.78 |
| 1YJD | 1AH1 | A | 30.70 | 0.77 |
| 2ARJ | 4XMN | F | 26.26 | 0.75 |
| 2VXQ | 1N10 | A | 41.30 | 0.89 |
| 2VXT | 4XFS | A | 94.23 | 0.93 |
| 2XQB | 2PSM | A | 69.91 | 0.89 |
| 3D9A | 2EQL | A | 49.22 | 0.93 |
| 3HI1 | 2BF1 | A | 33.94 | 0.83 |
| 3L5X | 3BPO | A | 99.05 | 0.83 |
| 3MXW | 2IBG | B | 70.00 | 0.92 |
| 3QWO | 1EDK | A | 50.94 | 0.86 |
| 3RKD | 3RKC | A | 88.19 | 0.49 |
| 4DN4 | 3FPU | B | 41.67 | 0.88 |
| 4DW2 | 2ODQ | A | 25.94 | 0.85 |
| 4ETQ | 2ZNC | A | 30.56 | 0.94 |
| 4G3Y | 1TNR | A | 36.43 | 0.86 |
| 4G6J | 3NJ5 | A | 35.37 | 0.85 |
| 4I3S | 2B4C | A | 61.96 | 0.82 |
| 4JZJ | 4RS1 | A | 31.97 | 0.33 |
| 4KI5 | 4QDR | A | 44.97 | 0.93 |
| 4L5F | 2HG0 | A | 45.92 | 0.80 |
| 4LVH | 5BNY | A | 40.89 | 0.90 |
| 4M62 | 4GQX | A | 23.94 | 0.81 |
| 4NP4 | 2GJ6 | A | 35.86 | 0.67 |
| 4RGO | 5FKA | C | 34.23 | 0.96 |
| 5D96 | 3G6O | A | 80.77 | 0.96 |
| Average | | | 46.13 | 0.82 |
| STD | | | 21.06 | 0.13 |

DOI: https://doi.org/10.7554/eLife.29023.024

effects of random variation, the process was repeated 1000 times for each target. On average, the success rates of plans using random designs were approximately 60%, compared to 85–88% for plans guided by docking (*Figure 4—figure supplement 2*). Random plans sufficed for some very small proteins (e.g., 3QWO, with 48 surface residues), yielding success rates reaching nearly 90%. On the other hand, for the moderately-sized 4KI5 (108 surface residues), EpiScope specified that it needed just a single design, which was indeed successful, though the random design approach success rate for this target was only 10%. In general, the substantially higher and consistent

performance of docking-guided design, along with the guidance it provides regarding the number of variants that must be tested in order to cover all the hypotheses, demonstrates that docking does indeed provide valuable information about where and how to target mutations.

## Mutations per design

To assess the trade-off between efficiency and precision of epitope localization, the number of mutations permitted per design was varied from 1 to 4 (*Figure 4—figure supplement 3*). With more mutations per design, fewer variants were required to sufficiently interrogate all docking models (*Figure 4—figure supplement 3A*), and an average of one design overlapped the true Ab epitope regardless of the number of mutations permitted (*Figure 4—figure supplement 3B–C*). However, the precision with which epitopes could be localized also decreased with increasing numbers of mutations per design (*Figure 4—figure supplement 3D–E*; Kendall's τ: 0.37 and 0.30 for the number of docking models and residues respectively). Most significantly, the success rate of epitope localization was highest when using three mutations/design (*Figure 4—figure supplement 2F*), suggesting that the choice of triple mutants for the prospective application provided a good balance between low relative experimental effort, acceptable epitope resolution, and high success rate.

## Inter-mutation distance

We next explored the relationship between allowed inter-mutation distance vs. resolution and success rate, since closer mutations could lead to more precise identification of the epitope but at the expense of actually hitting epitopes less frequently. We found that average Cα distances in our initial designs were generally 11 to 15 Å (*Figure 4—figure supplement 4A*). Double and triple mutation variants were then designed while systematically varying the distance cut-off in order to obtain different average inter-mutation distances. Single mutation variants were also designed in order to provide a baseline for comparison. For ease of interpretation in terms of the effects of the distance threshold, the algorithm was constrained to generate a single design of 1, 2, or 3 mutations with which to disrupt docking models.

The baseline success rate in hitting an epitope with a single optimized mutation was approximately 25% (*Figure 4—figure supplement 4B*); for reference, random single mutations hit epitopes about 14% of the time (not shown in the figure). Increasing the mutational load had a substantial impact, with the success rate jumping to 50% and 55% for just one double or triple mutant (*Figure 4—figure supplement 4B–D*). Spreading mutations out more than the initially selected 12 Å average did not improve the success rate, perhaps due to lack of coherence. Bringing them too close together likewise decreased the success rate, though we note that this observation is limited by the fact that there were not many designs available at shorter cut-offs (particularly 6 Å). Filtering docking models according to consistency with disruptive mutations left an average of about 12 models covering about 47% of the Ag surface in the baseline case of a single optimized mutation (an average of 5.6 docking models spanning 32% of the Ag surface for random single mutations, not shown in the figure). Double and triple mutants resulted in a few more models covering slightly more surface residues (16 and 17 models, and 52% and 56% of the surface at 12 Å), not sacrificing much in resolution in order to obtain their better success rates. Thus, while the cut-off did result in a trade-off between the resolution and the success rate, a 10–12 Å threshold seemed to provide the best balance.

## Epitope definition

Finally, we considered the impact of the definition of 'ground truth' on the assessment of these retrospective tests. While the epitope definitions used so far were those deposited in the IEDB as experimentally verified, the larger set of 'binding interface' residues could also be considered. We analyzed our results in terms of such residues, as determined by an inter-heavy atom distance of 5 Å in the co-crystal structure. *Table 6* details that, as would be expected, this broader definition yields improved hit rates, with only three targets missed using either crystal structures or homology models (compared to 4 and 5, respectively, for IEDB epitopes). As with the IEDB specification of epitopes, these additional results suggest that both crystal structures and homology models of the Ags may be sufficient to localize Ab:Ag binding. While mutations at binding interface positions may not

**Table 6.** Success rates with epitopes defined according to IEDB or according to contacts in the binding interface.
Success is indicated as 'T' and failure as 'F'. In test cases colored blue, EpiScope failed to find IEDB epitopes but did find binding interface residues.

| Target | Crystal structure | | Model structure | | Target | Crystal structure | | Model structure | |
|---|---|---|---|---|---|---|---|---|---|
| | IEDB | Interface | IEDB | Interface | | IEDB | Interface | IEDB | Interface |
| 1FE8 | T | T | F | F | 3QWO | T | T | T | T |
| 1FNS | T | T | T | T | 3RKD | T | T | T | T |
| 1H0D | F | F | T | T | 4DN4 | T | T | T | T |
| 1LK3 | T | T | T | T | 4DW2 | T | T | T | T |
| 1OAZ | F | F | F | F | 4ETQ | T | T | T | T |
| 1OB1 | T | T | T | T | 4G3Y | T | T | T | T |
| 1RJL | T | T | T | T | 4G6J | T | T | T | T |
| 1V7M | T | T | T | T | 4I3S | T | T | T | T |
| 1YJD | T | T | T | T | 4JZJ | T | T | T | T |
| 2ARJ | T | T | T | T | 4KI5 | T | T | T | T |
| 2VXQ | T | T | T | T | 4L5F | T | T | T | T |
| 2VXT | T | T | T | T | 4LVH | T | T | F | T |
| 2XQB | T | T | T | T | 4M62 | T | T | F | T |
| 3D9A | T | T | T | T | 4NP4 | T | T | T | T |
| 3HI1 | F | F | F | F | 4RGO | F | T | T | T |
| 3L5X | T | T | T | T | 5D96 | T | T | T | T |
| 3MXW | T | T | T | T | Total | 29 (88%) | 30 (91%) | 28 (85%) | |

DOI: https://doi.org/10.7554/eLife.29023.025

completely disrupt Ab-Ag binding, they may be good enough to enable detection of binding reduction, particularly since EpiScope selects highly-disruptive mutations.

## Computationally-driven epitope binning and localization of multiple Abs targeting the same Ag

As observed with B7H6, computational modeling and design can uncover similarities and differences in possible binding modes of multiple different Abs against the same Ag, enabling design of a panel of variants to simultaneously map all of the epitopes. We sought to evaluate how this performance would scale for a larger set of Abs. We considered 12 immunization-induced Abs previously found to target four epitopes (*Figure 5A*) on the D8 envelope protein of vaccinia virus (*Sela-Culang et al., 2014*; *Matho et al., 2014*), the active component in smallpox vaccines. This retrospective test set thus serves as proof of concept for extracting relevant binding information from Ab sequences in order to characterize humoral responses to immunization, while also mirroring Ab discovery/isolation efforts where multiple Abs against a single Ag are isolated at once.

The docking models generated for the 12 different anti-D8 Abs were fairly indistinguishable (*Figure 5B*); in fact, the models for each Ab covered on average ~80% of the surface of the Ag (*Figure 5—figure supplement 1*), leaving little room for differentiation. However, quite strikingly, similarities between EpiScope-generated variants designed to disrupt the docking models revealed patterns among the Abs (*Figure 5C*) that were similar to those observed in experimentally determined competitive binding assays (*Figure 5A*). Thus the combination of docking and design elucidated amino acid level patterns of specificity driving Ab–Ag interactions. It bears noting that EpiScope was still able to identify epitope residues: in the crystal structure for the D8-LA5 (PDB ID 4ETQ), two designs are in the LA5 Ab binding regions (*Figure 5—figure supplement 2*).

Since Ab recognition of Ag is driven by the Ab complementarity determining regions (CDRs), Abs with very similar CDR sequences would be expected to have very similar epitopes. To explore the impact of CDR sequence similarity on the relationship among docking, disruptive design, and binning, we selected one Ab for each of the seven unique sets of CDR sequences represented among

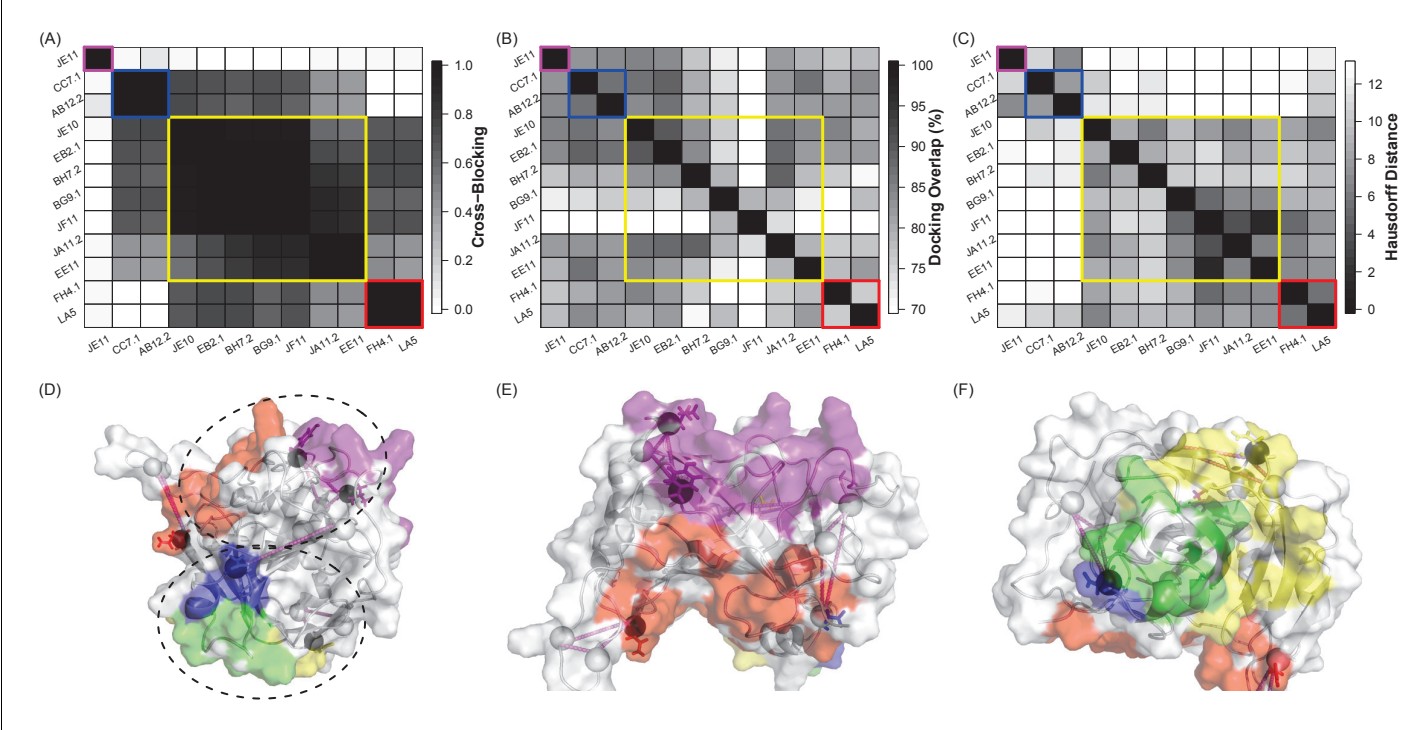

**Figure 5.** A small set of Ag variants has the potential to simultaneously localize multiple Ab epitopes for a single Ag. (**A**) Heat map of competitive binding data (*Sela-Culang et al., 2014*) for 12 antibodies directed against the vaccinia virus D8 protein, with the extent of cross-blocking ranging from 0.0 (white, no effect) to 1.0 (black, complete blocking). Colors in all panels refer to the four Ab groups identified by this competition assay (I: purple, II: blue, III: yellow, and IV: red). (**B**) Heat map of the overlap between ClusPro-generated docks for each pair of Abs, ranging from 60% (white) to 100% (black). (**C**) Heat map of the average Hausdorff distance between Ag variants designed for each Ab, ranging from 0 (identical mutation sites, black) to 12 (white). (**D–F**) Ag variants designed to disrupt one Ab from each group (I: JE11, II: CC7.1, III: EE11, IV: LA5) are represented as triangles. Four designs were sufficient to cover all docking models, and the designs overlapped all of the epitope groups. True epitopes are color coded by group on the surface of the antigen; epitopes in group II and III overlapped, and are colored in green. Design residues overlapping the true epitopes are indicated with circles. (**E and F**) Zoomed views of epitope faces.

DOI: https://doi.org/10.7554/eLife.29023.026

The following source data and figure supplements are available for figure 5:

**Source data 1.** Raw data values for *Figure 5A, B and C*.
DOI: https://doi.org/10.7554/eLife.29023.030

**Figure supplement 1.** ClusPro generates docking models that cover nearly all surface residues for the set of 12 VACV anti-D8 envelope targeting Abs.
DOI: https://doi.org/10.7554/eLife.29023.027

**Figure supplement 1—source data 1.** Raw data values for *Figure 4—figure supplement 1*.
DOI: https://doi.org/10.7554/eLife.29023.031

**Figure supplement 2.** Identification of vaccinia D8 epitopes against LA5.
DOI: https://doi.org/10.7554/eLife.29023.028

**Figure supplement 3.** Heatmaps of sequence identity between the selected 7 VACV anti-D8 Envelope targeting Abs.
DOI: https://doi.org/10.7554/eLife.29023.029

**Figure supplement 3—source data 2.** Raw data values for *Figure 5—figure supplement 3*.
DOI: https://doi.org/10.7554/eLife.29023.032

the 12 Abs (yielding JE11, AB12.2, BG9.1, EB2.1, EE11, JE10, and FH4.1). The experimental epitope binning results for this subset of Abs largely reflected heavy chain CDR (CDR-H) sequence similarity, and did not significantly depend on light chain CDRs (*Figure 5—figure supplement 3*). Thus we may conclude that for this set of Abs, the CDR-Hs largely drive binding. However, without such experimental data, it is not obvious how important each CDR is to the binding profile; e.g., there are certainly cases where light chains are very important for strong binding (*Ko et al., 2015*). Unfortunately, simply comparing overall CDR sequence similarity ('all' in *Figure 5—figure supplement 3*), as might be appropriate without any assumptions, yields a pattern that doesn't reflect binning as

well as that for individual CDR-Hs. More generally, it is not easy to predict how much variation in an individual CDR sequence will impact binding, or how to combine variation across multiple CDRs to assess an overall effect. On the other hand, EpiScope naturally integrates this sequence information into structural models, predictions of possible binding modes, and design of disruptive mutations. Thus EpiScope's 'binning' pattern does reflect the experimental binning results.

With Abs binned into four separate groups, a panel of Ag variants could be designed to localize the epitope of each group based on a representative member; here, for the sake of testing we used the Ab from each group that had previously been structurally characterized (*Matho et al., 2014*). While a total of 18 designs would be required to localize each Ab independently (JE11: 5, CC7.1: 4, EE11: 4, LA5: 5), a multi-Ab panel of only four variants could simultaneously cover all 116 docking models from all 4 Abs (*Figure 5D*). Remarkably, each design contained one or two mutations overlapping the characterized binding interface of the representative Ab from a particular binning group, suggesting that the variants localized to meaningful epitope regions and may further serve as epitope probes with which to characterize or select new Abs with varying specificities in the future. This investigation thus demonstrated that not only is it possible to obtain epitope grouping information without experimental effort, but also that subsequent experimental effort can be greatly reduced while localizing multiple Ab epitopes. Designed Ag variants could be of further utility as probes to profile epitope specificities of polyclonal serum samples and to investigate correlates of vaccine protection or efficacy.

## Purely computational prediction

To characterize potential results from purely computational epitope prediction, we applied to all of our targets a state-of-the-art predictor, the computational component of the integrated epitope localization method PEASE (*Sela-Culang et al., 2014*). Similar to EpiScope, PEASE accepts Ab sequences and Ag structures as inputs, but instead of explicitly docking Ab and Ag, it utilizes machine learning methods based on Ab-Ag binding interface characteristics to predict epitope 'patches' of 4–5 residues each. While the PEASE approach further incorporates the results of competition experiments to refine patch predictions for epitope-grouped antibodies, here we consider the accuracy of just the Ab-specific patches themselves. In characterizing PEASE results we used the 'residue-score' (RS) cutoff of 0.43, which was determined optimal in retrospective test cases (*Sela-Culang et al., 2014*), but we note that results were substantially affected by this value, adding a layer of complexity to the analysis. PEASE provides a ranking of patches, so to make a balanced comparison, we considered an equal number of top-ranked PEASE predictions to the number of variants designed by EpiScope.

For TZ47, none of the top 4 PEASE patches (covering 13 residues) contained residues proximal to the localized epitope, but for PB11 the top PEASE patch (five residues) contained at least two residues overlapping with mutations in disruptive chimera or EpiScope designs (*Table 7*). Over all of the retrospective test cases (*Table 8*), the top PEASE patches overlapped with epitope residues 52% of

**Table 7.** Comparison of residues predicted by PEASE for TZ47 and PB11 to mutations included in disruptive EpiScope designs. Residue score cut-off 0.43 was used for PEASE.

| Patch | Predicted patch residue positions | Patch score | Disruptive EpiScope design mutation positions |
|---|---|---|---|
| TZ47-Patch 1 | 158,159,160,161,162 | 0.41 | 154, 157, 217 (TZ47-Ag4) |
| TZ47-Patch2 | 158,160,161,162,163 | 0.4 | 154, 157, 216 (MULTI-4) |
| TZ47-Patch3 | 1,29,30,31,32 | 0.4 | |
| TZ47-Patch4 | 1,2,30,31,106 | 0.39 | |
| PB-Patch1 | 1,2,30,31,106 | 0.47 | 51, 52, 99 (PB-Ag2) |
| PB-Patch2 | 46,47,48,49,50 | 0.41 | 57, 84, 98 (MULTI-1) |
| PB-Patch3 | 158,160,161,162,163 | 0.4 | |
| PB-Patch4 | 195,196,197,198,203 | 0.38 | |
| PB-Patch5 | 123,124,125,126,139 | 0.38 | |

DOI: https://doi.org/10.7554/eLife.29023.033

**Table 8.** Comparison of predictive components of PEASE and EpiScope on retrospective test set of 33 non-redundant Ab-Ag pairs. The number of designs needed/considered indicates the number of designs generated by EpiScope to cover all ClusPro docking models. An equivalent number of the top ranked PEASE patch predictions are considered for each Ab. Coloring highlights the cases in which Episcope (green) or PEASE (red) succeeded where the other method failed. Grey coloring indicates cases in which both methods failed.

| | Crystal structure of ag | | | Modeled structure of ag | | |
|---|---|---|---|---|---|---|
| Target | # of Designs Needed/ Considered | # of EpiScopeDesigns Overlapping True Epitope | # of PEASE patches Overlapping True Epitope | # of Designs Needed/ Considered | # of EpiScopeDesigns Overlapping True Epitope | # of PEASE patches Overlapping True Epitope |
| 1FE8 | 4 | 2 | 4 | 3 | 0 | 0 |
| 1FNS | 5 | 2 | 5 | 2 | 1 | 2 |
| 1H0D | 3 | 0 | 0 | 2 | 1 | 0 |
| 1LK3 | 3 | 1 | 0 | 3 | 1 | 0 |
| 1OAZ | 2 | 0 | 2 | 2 | 0 | 2 |
| 1OB1 | 3 | 1 | 0 | 4 | 2 | 4 |
| 1RJL | 3 | 1 | 3 | 2 | 1 | 2 |
| 1 V7M | 6 | 1 | 0 | 3 | 1 | 2 |
| 1YJD | 2 | 1 | 0 | 3 | 1 | 0 |
| 2ARJ | 3 | 1 | 3 | 3 | 1 | 3 |
| 2VXQ | 3 | 1 | 3 | 3 | 1 | 3 |
| 2VXT | 3 | 1 | 3 | 3 | 1 | 3 |
| 2XQB | 2 | 1 | 2 | 3 | 1 | 3 |
| 3D9A | 3 | 1 | 0 | 2 | 1 | 0 |
| 3HI1 | 4 | 0 | 0 | 3 | 0 | 0 |
| 3L5X | 6 | 1 | 6 | 3 | 2 | 3 |
| 3MXW | 2 | 1 | 2 | 2 | 1 | 2 |
| 3QWO | 3 | 2 | 3 | 4 | 3 | 4 |
| 3RKD | 5 | 2 | 3 | 3 | 1 | 2 |
| 4DN4 | 2 | 1 | 0 | 2 | 1 | 0 |
| 4DW2 | 4 | 1 | 2 | 3 | 1 | 1 |
| 4ETQ | 4 | 1 | 1 | 3 | 1 | 1 |
| 4G3Y | 3 | 1 | 0 | 4 | 2 | 3 |
| 4G6J | 3 | 1 | 2 | 3 | 1 | 3 |
| 4I3S | 4 | 2 | 3 | 4 | 1 | 2 |
| 4JZJ | 4 | 1 | 0 | 6 | 4 | 0 |
| 4KI5 | 1 | 1 | 0 | 1 | 1 | 0 |
| 4 L5F | 2 | 1 | 0 | 3 | 1 | 0 |
| 4LVH | 5 | 1 | 0 | 6 | 0 | 2 |
| 4 M62 | 2 | 1 | 0 | 2 | 0 | 0 |
| 4 NP4 | 3 | 1 | 0 | 5 | 3 | 0 |
| 4RGO | 3 | 0 | 2 | 6 | 1 | 3 |
| 5D96 | 4 | 1 | 0 | 4 | 1 | 0 |

DOI: https://doi.org/10.7554/eLife.29023.034

the time (compared to 88% with crystal and 85% with homology model for EpiScope designs). EpiScope succeeded in 14 cases where PEASE failed and PEASE succeeded in two cases where EpiScope failed. When the computation-only portion of PEASE was applied to the set of 12 VACV Abs, only ~40% of residues contained within the combined set of top predicted patches were part of Ab

epitopes (*Sela-Culang et al., 2014*). When considering the top ranked PEASE patch prediction for each of the structurally characterized four representative Abs, only two epitopes were localized, one correctly (Group I epitope predicted for Group I Ab JE11) and one fortuitously (Group IV epitope predicted for group II Ab CC7.1). Furthermore, the same top patch prediction was returned for 3 out of the 4 Abs. However, residues comprising the binding interface overlapped with at least one of the top 11 PEASE patch predictions for each Ab, suggesting a potential benefit to incorporating PEASE predictions into the generation of hypotheses for EpiScope-directed experimental validation, focusing experimental effort on those patches that PEASE identifies as most important, either based on purely computational experimental analysis or by integration of prior experimental data.

## Discussion

We have demonstrated that the combination of computational docking and computational protein design can explicate binding information implicitly encoded in Ab sequence and thereby drive efficient localization of epitopes. To our knowledge, EpiScope is the first method to directly optimize experimental validation of in silico epitope predictions, designing rich combinations of mutations in Ag variants to disrupt predicted Ab binding. We successfully localized epitopes prospectively for two Abs using only Ab sequence and Ag structure as inputs to the design process. This work thus significantly elaborates the recent push for integrated computational-experimental epitope mapping, which previously required initial incorporation of experimental data from competitive epitope binning assays (*Sela-Culang et al., 2014*) or neutralization assays of viral variants (*Chuang et al., 2013*) in order to generate a ranked list of predicted epitopes for experimental evaluation. In contrast, we show that starting with only sequence information, comprehensive experimental testing can be optimized to cover all epitope predictions.

Retrospective analysis bolstered the generality of the lessons from the successful prospective application, demonstrating an expected 88% success rate with only about three experiments – a high likelihood of successful epitope localization with minimal experimental effort. Notably, the design process itself determines how many experiments are required to test all the computational hypotheses. While docking itself was not sufficient to confidently identify epitopes, the information it provided was critical in driving the experiments, as performance suffered when using random size-matched designs instead of docking-based ones. In expansion to multiple Abs against a single Ag, the computational analysis integrated information about sequence and structural similarity into hypotheses about binding similarity, and by itself was sufficient to epitope bin 12 Abs targeting four different epitopes on a single Ag. By leveraging commonalities of putative epitopes, multi-Ab-targeting sets of Ag variant designs were comparable in size to those for single Abs, and were indeed sufficient to localize all epitopes simultaneously in both retrospective and prospective cases. Thus, this methodology may be useful for the translation of high-throughput Ab repertoire sequencing data into the identification and definition of clinically relevant epitopes. The ability to epitope bin in silico and localize multiple epitopes through the rational design of optimal Ag variant panels offers the potential to incorporate epitope diversity earlier into Ab development pipelines.

The integrated computational-experimental approach thus employs computational modeling to extract and exploit crucial features of molecular recognition encoded in the amino acid sequences of the Ab and Ag. By computationally focusing experimental effort, it offers potentially significant time and cost savings relative to purely experimental evaluation methods, and potentially significant accuracy improvements relative to purely computational prediction methods. It also strikes a balance between the relatively high-resolution localization provided by comprehensive experimental studies, and the relatively low-resolution information provided by high-throughput competition studies. We now discuss some of the impacts of relying on computation, the niche filled relative to experimentally-driven efforts, and the outlook for future developments and studies.

### Limitations
In general, computationally-driven methods critically depend on the quality of inputs provided, the degrees of freedom allowed, and the algorithms employed. These factors impact both what is possible with computationally-driven epitope mapping and how well it performs in different scenarios.

## Ab homology models

With the ever-increasing number of solved crystal structures, Ab homology modeling approaches routinely achieve Angstrom-level accuracy for framework regions and CDRs other than H3, for which state of the art is typically 1.5 ~ 3 Å (*Marks and Deane, 2017*). This level of accuracy was obtained here, despite limiting template identity, and consequently Ab model quality was not a major driving factor for success (*Figure 6—figure supplement 1A and B*). In fact, the best modeled Ab structure, 1FE8 (CDR-H3 RMSD: 0.63 Å), failed perhaps due to poor docking ($f_{nat}$ with the crystal Ag structure: 0.1, just above 'acceptable'), whereas epitope overlapping positions were identified using the worst Ab model, 2XQB (CDR-H3 RMSD: 7.21 Å and $f_{nat}$: 0.32, 'medium'). In settings where Abs are harder to model (e.g., antibodies with very long CDR-H3s, or post-translationally modified CDRs), performance may suffer. While in theory the approach presented here may also apply equally well to alternative formats from other species and antibody-mimetics, in practice it depends on the quality of resulting models. It is possible that performance may be even better, for formats with more-constrained and more-easily-modeled binding regions.

## Ag homology models

As with Abs, while modeling success on any specific Ag target depends very much on the availability of high-quality, well-matched templates, the continued expansion of structural databases and improvements in modeling algorithms have led to fairly routine Angstrom-level models (*Moult et al., 2016*). Here, even while again attempting to use only moderately-similar templates, the models were uniformly of high quality, and the quality did not appear to play a significant role in the results (*Figure 6—figure supplement 1C and D*). For example, the Ag structure of 4RGO was accurately predicted but failed, again perhaps due to poor docking ($f_{nat}$: 0.16). However, the worst model, 4JZJ, still yielded a successful design possibly because the binding interface was still modeled sufficiently accurately to support docking ($f_{nat}$: 0.31). While not observed here, if, compared to the model, the Ag undergoes substantial conformational changes affecting the epitope region, or if post-translational modifications interfere with (or even comprise) the epitope, epitope localization results may certainly suffer.

## Ab:Ag docking models

As discussed in the introduction, docking generally produces an acceptable-quality model among the top set (*Brenke et al., 2012*). Furthermore, docking tends to perform better with crystal structures than with homology models (*Rodrigues et al., 2013*). We observed both phenomena here (*Figure 6*). This analysis also showed that docking model quality was the major factor driving success or failure of the approach. EpiScope was always able to identify epitopes for targets with good docking models, i.e., those above 'medium' according to the CAPRI definition (*Lensink et al., 2007*). While the failure cases all had poor docking models (e.g., *Figure 6—figure supplement 2A*), EpiScope did sometimes succeed even in cases with poor docking models (4I3S; $f_{nat}$: 0.05). With continued improvement of docking methods, the 'medium' case may become the norm, but for any particular target, docking may fail; this is a particular risk if there are poorly modeled portions of the Ab or Ag, or if there is substantial conformational change upon binding.

In addition to the quality of docking models, their number and diversity will also affect the results, as more variants may be required to cover more diverse models. By using ClusPro here, each target had only a relatively small (around 30) and diverse set of docking models clustered from a much larger initial set. There was still some correlation observable between the number of docking models and the number of final selected designs: a correlation coefficient (Kendall's $\tau$) of 0.397 for Ag crystal structures, and 0.443 for Ag homology models.

## Ag mutations

Sets of mutations must be chosen to disrupt Ab binding while preserving Ag stability. While in general modeling the effects of mutations on binding and stability remains very challenging, the case here is perhaps the most benign scenario for both, in that we seek only to disrupt binding (not improve it) and maintain stability (not improve it) while mutating solvent exposed (not core) residues that are fairly well spread apart (typically not directly interacting). Thus, much as with alanine scanning, the chosen mutations can be expected to be relatively benign. For the prospective application,

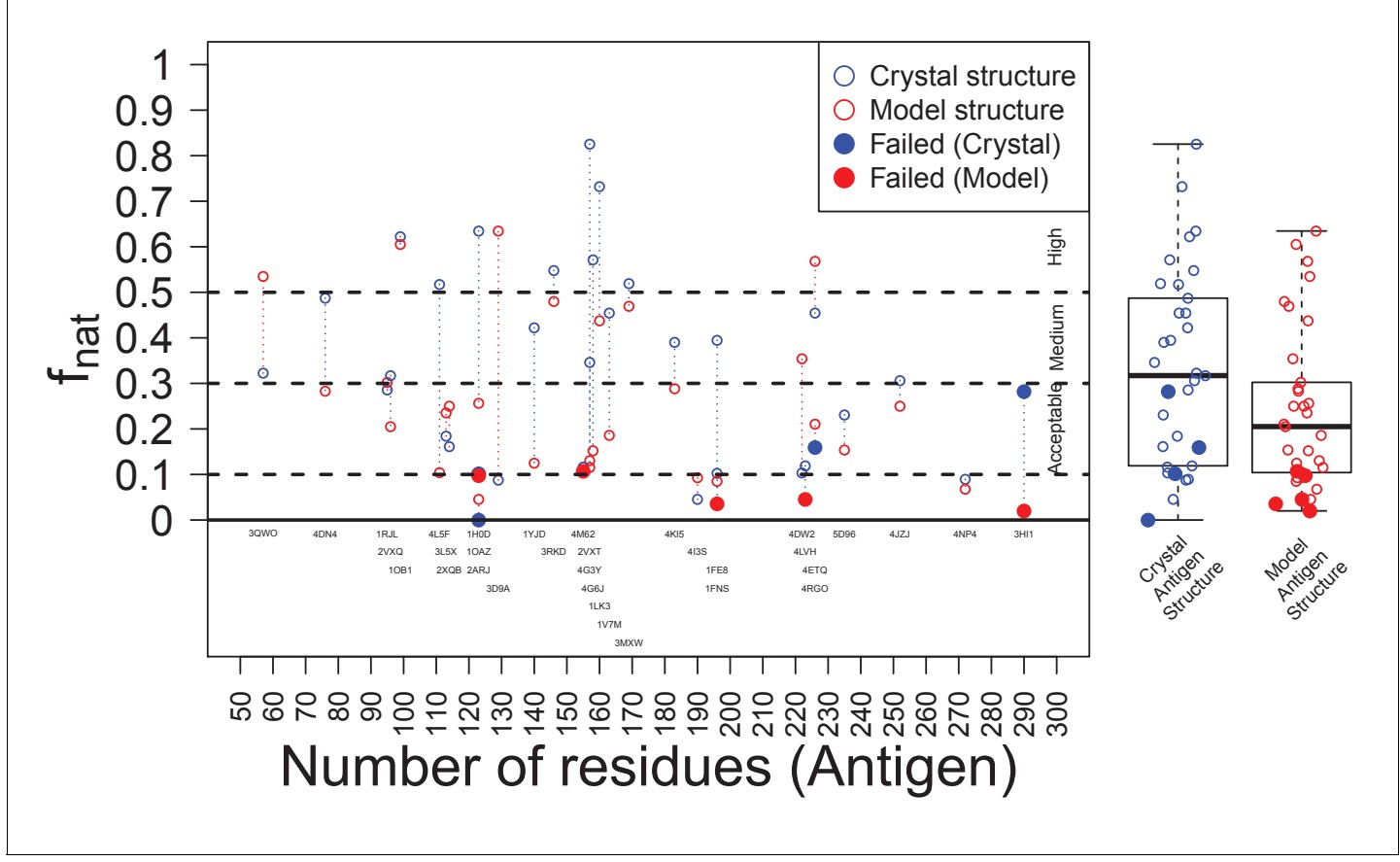

**Figure 6.** Success of EpiScope and the quality of docking models. In general, docking using Ag crystal structures is better than using Ag homology models according to the $f_{nat}$ value; it is above 'medium' for crystal structures but only 'acceptable' for model structures. Poor docking models are necessary, but not sufficient, for the failure of the EpiScope approach: EpiScope still identifies epitopes for some poorly docked models, but all failed cases have low $f_{nat}$ values.

DOI: https://doi.org/10.7554/eLife.29023.035

The following source data and figure supplements are available for figure 6:

**Source data 1.** Raw data values for *Figure 6*.
DOI: https://doi.org/10.7554/eLife.29023.006

**Figure supplement 1.** Examples of success and failure depending on qualities of Ab structure (**A and B**) and Ag structure (**C and D**).
DOI: https://doi.org/10.7554/eLife.29023.036

**Figure supplement 2.** Two examples of EpiScope failure cases.
DOI: https://doi.org/10.7554/eLife.29023.037

we showed that indeed the designed mutations tested did not destabilize the Ag in terms of eliminating its ability to bind its natural ligand. While we could not explicitly evaluate that for the retrospective studies, the molecular modeling method employed is one of many similar well-established techniques for predicting and designing for stability and has been successfully applied to other challenging cases such as mutation of hydrophobic core residues in T cell epitope deletion (*Blazanovic et al., 2015*; *Salvat et al., 2015*; *Zhao et al., 2015*).

By restricting mutations to those appearing among homologs, we leverage nature's experiments to improve the chance of success, though at the cost of reducing the degrees of freedom to consider. This appears to have caused one of the failures, when the lack of homologous sequence information for portions of the Ag limited the mutational choices considered (*Figure 6—figure supplement 2B–C*). Alternative approaches could leverage structural modeling to fill in such gaps and even to expand the mutations considered throughout based on initial individual energy evaluations. Sequence and structural modeling could even be integrated in a Pareto optimal fashion (*He et al., 2012*) to balance reliance on both sources of information.

## Computational cost

Given docking models, the design process proceeds through several steps, with the most computationally intensive being designing sets of disruptive mutations for each docking model independently, and clustering these designs to cover all models simultaneously. We used here a design algorithm based on integer linear programming, thereby generating provably optimal global designs rather than stochastically sampling. This guaranteed optimization approach comes at some computational cost, but given the relatively few degrees of freedom here, required less than a day per target on using 10 nodes on a cluster. The selection of designs covering all models was performed by a K-medoids clustering script followed by exhaustive testing of combinations across clusters. Both steps could be further optimized or optimality could be traded for efficiency, but the time required for computation is already much cheaper than that required for experiment.

## Resolution

While experimental approaches, from traditional alanine-scanning mutagenesis to advanced techniques including the combination approach of comprehensive mutagenesis libraries and deep sequencing (*Kowalsky et al., 2015*), can provide unparalleled residue-level detail, they can require significantly more experimental effort than the computationally-directed approach. In addition, mutation of non-epitope residues can confer phenotypic changes in binding (*Guinto et al., 1999*; *Dang et al., 1997*), resulting in false assignment of these residues to the binding interface (*Greenspan and Di Cera, 1999*). EpiScope attempts to address these limitations by evaluating and optimizing the potential of individual residue changes to disrupt stability and Ab binding in predicted docking models, although there theoretically remains the potential for EpiScope-designed mutations to similarly misrepresent epitopes. Fine-grained epitope characterization is often a goal during late stages of Ab development after the number of promising candidates has been narrowed down to a manageable number for such time- and cost-intensive efforts. However, consideration of epitopes earlier during initial large-scale screens may enable selection for epitope diversity, and may be better served by the more efficient and less resource intensive, albeit less-detailed, characterization afforded by Episcope.

Chimeragenesis provides an alternative method to incorporate multiple potentially Ab-disrupting but stability-preserving mutations. However, designing suitable chimeras is difficult as is the interpretation of binding assay results, due to the complex relationship between linear recombination and spatial organization of epitopes. Anecdotally, we had tested eight chimeric variants of macaque and human B7H6 homologs in an attempt to localize the TZ47 epitope, but failed to do so before designing a 9[th] chimera (in Results) based on the EpiScope-designed disruptive variant. We note that the computational design was 'turn key' based on only Ab sequence and Ag structure and succeeded with the promised number of experiments. The designed Ag variants also contained fewer mutations on average than chimeras, providing greater resolution on the binding epitope and potentially greater expression fidelity.

EpiScope balances the level of detail obtained in epitope localization with the experimental effort required, using only a few variants but leaving the epitope only roughly defined. It provides additional docking information that is not captured by standard mutagenesis-based methods, of particular use for those seeking to engineer the epitope or paratope to create enhanced reagents. On the other hand, no single computationally generated docking model is necessarily correct (recall average $f_{nat}$ = 32%), so follow-up experiments would be necessary to achieve the level of resolution provided by experimentally driven efforts discussed above. As demonstrated by retrospective cases, EpiScope can significantly decrease the amount of subsequent effort required by filtering relevant Ag surface residues to the vicinity of affected computational docks. Results of focused experiments, such as alanine scanning, may then be used to further filter/predict the most native-like docking model. Alternatively, a subsequent computationally-driven round of targeted mutations could be optimized. Docking models would be concentrated on the region, disruptive mutations designed, and then variants selected so as to expand the set of identified epitope residues and more finely discriminate among the models. In summary, EpiScope requires minimal prior knowledge and experimental effort and most efficiently ensures the successful localization of Ab epitopes given Ab sequence and Ag structure as inputs to the design process.

## Outlook

Future uses of computationally-directed epitope mapping may enable high-throughput Ab epitope binning and localization using a minimal set of Ag variants for large panels of Abs, offering opportunities for the profiling of polyclonal serum samples in various disease settings and/or earlier selection for epitope specificity in Ab discovery pipelines (*Brooks et al., 2014*). Collectively, such high-throughput epitope characterization combined with rapidly advancing B-cell isolation and NGS technology may enable insights into the development of humoral immunity contributing to health/disease, including investigations of which epitopes more/less commonly elicit Abs, are targeted by Abs at different stages of disease, correlate with clinical status, etc. Investigations of such critical questions could then inform immunogen design and vaccine strategies, or the de novo design of therapeutic Abs targeting functionally relevant epitopes to enable novel mechanisms of action. In summary, EpiScope utilizes Ab sequence-encoded binding information to offer a highly efficient epitope localization strategy to keep up with rapidly advancing Ag-specific B cell sorting and next generation sequencing efforts and offers the exciting potential to advance early Ab discovery and development efforts, evaluation of humoral responses in various disease/vaccination settings, and rational epitope-focused vaccine design.

# Materials and methods

## Computational method: EpiScope

As summarized in *Figure 1*, the computational design component of the EpiScope protocol generates representative Ab:Ag docking models, designs Ag variants so as to disrupt the various models, and selects a small set of those variants so as to ensure all (or as many as possible) of the docking models will be disrupted by at least one variant. These steps were instantiated here as follows.

Representative docking models were generated from the ClusPro webserver (*Comeau et al., 2004b*) in Ab mode with non-CDR masking (*Brenke et al., 2012*). All returned models were treated equally for subsequent analysis; depending on the target, this included from 13 to 30 representative docking models for the retrospective test sets (*Figure 1A*).

Ag variants were designed by a customized version of the EpiSweep protein redesign algorithm (*Choi et al., 2017*; *Parker et al., 2013*), modified to delete predicted Ab epitopes instead of predicted T cell epitopes (*Figure 1B*). Briefly, the protein design method selected from 1 to 4 mutations per Ag design predicted to be disruptive of Ab binding according to a docking model while simultaneously ensuring that the mutations would not be detrimental to Ag stability. Binding disruption was predicted according to the INT5 statistical potential (*Pons et al., 2011*), part of the SIPPER scoring function shown to be successful in protein docking benchmarks (*Pons et al., 2011*; *Moal et al., 2013*). A rotameric energy was used to predict effects of mutation on stability (*Pearlman et al., 1995*; *Chen et al., 2009*; *Gainza et al., 2012*). In order to restrict mutations to those most likely to be acceptable for Ag stability, a homology-based filter was employed (*Choi et al., 2017*; *Parker et al., 2013*), considering only evolutionarily-accepted variations appearing in homologous sequences found within 3 iterations of PSI-BLAST (e-value <0.001) (*Altschul et al., 1997*). Amino acids that were not predicted to disrupt any of the docking models (disruptive potential score <0) were removed from the list of choices.

After the generation of sets of mutations for each docking model, further filtration steps were performed. Any design whose rotameric energy was worse than the wild type was excluded, as was any whose mutations had average Cα distances >12 Å. Designs often overlapped with multiple docking models; only designed mutations that were disruptive to all of the docking poses in contact were included. In the case of identical positions with different mutations, the one with the most disruptive binding score was considered.

All remaining designs were then clustered using the *K*-medoids algorithm with the Hausdorff distance for the Cα coordinates of mutated positions. Intuitively, the Hausdorff distance assesses the distance between two sets of points by finding for each point in one set the closest neighbor in the other, and identifying the furthest neighbors. More precisely, if $A$ has $n$ points $\{a_1, a_2, \ldots, a_n\}$ and $B$ has $m$ points $\{b_1, b_2, \ldots, b_m\}$, then the Hausdorff distance $h(A,B)$ between $A$ and $B$ is

$$h(A,B) = \max\left\{ a \in A \max\{b \in B \min d(a,\ b)\},\ b \in B \max\left\{ \min_{a \in A} d(b,\ a) \right\} \right\}$$

*K*-medoids clustering was implemented using a script from the pyclust package (version 0.1.3, obtained from PyPI), modified to employ Hausdorff distance. *K* was started at one and was increased until any combination of designs, one from each cluster, was found to disrupt all of the docking models (*Figure 1B*). The set of designs with the most disruptive binding score was selected as the final set (*Figure 1C*). An initial implementation of EpiScope used for the prospective application employed an alternative approach to variant set selection, starting with a 'centroid' of all designs with addition of variants to maximize coverage of all docking models. Designs for both the prospective targets generated using the current implementation included disruptive mutations proximal to those generated with the previous implementation and validated experimentally in the results.

## Structure preparation for B7H6

The crystal structure of unbound B7H6 was previously determined (PDB code: 3PV7) at 2.0 Å. There is a missing loop in the crystal structure (Chain A 150–157; DQVGMKEN). The loop was modeled using FREAD (*Choi and Deane, 2010*; *Kelm et al., 2014*) and a loop from 1O57:A 253–260 (STIN<u>M-KE</u>K) was found to be the best match. Anchor residues of the grafted loop (including backbone atoms) were minimized using the Tinker molecular dynamics package (*Ponder, 2004*)(ver. 6) using AMBER99sb (*Hornak et al., 2006*) with the GB/SA implicit solvent model (*Still et al., 1990*) while keeping the overall loop structure.

TZ47 is a murine Ab that binds to human B7H6. Its structure is unknown and thus the homology model from a previous study (*Choi et al., 2015*) was used. A model for PB11 was created using the PIGS modeling server (*Marcatili et al., 2015*; *Marcatili et al., 2014*). The model structure including backbone atoms was minimized using Tinker as described above. ClusPro in Ab mode with CDR masking generated 28 docking poses for each Ab.

## Retrospective test set

Thirty-three Ab-Ag pairs with complex structures solved by X-ray crystallography (*Table 3*) were selected from SAbDab (*Dunbar et al., 2014*) according to the following criteria: pairwise Ab sequence identity <70%, pairwise Ag sequence identity <70%, resolution <3 Å, single-chained, and >50 and <300 residues in length. Structures missing any backbone atoms were excluded.

In order to consider practical applications in a realistic setting, all Ab structures were homology modeled in a manner established to simulate 'hard-to-model' situations, for which template sequence identity is less than 90% (*Fasnacht et al., 2014*). Abs were modeled using the PIGS webserver with restricted sequence identity templates (<80% to the target) followed by side chain energy minimization using Tinker as described above. The quality of the Ab models was extremely accurate (*Table 4*), both overall (TM-score: 0.95) and for CDRs (<2 Å for CDR-H3 and sub-Angstrom for others), consistent with a report on CDR-H3 modeling quality (*Choi and Deane, 2011*).

Ag targets were either crystal structures from the bound complexes, or homology models built by SWISS-MODEL (*Biasini et al., 2014*) with default parameters applied to templates. Again, in order to represent realistic application, templates were selected to keep sequence identity low. When possible, the sequence identity was restricted to be less than 50%, but for nine cases had to be increased due to lack of any targets at that cut off. The resulting average template sequence identity was 46%, ranging from 23% to 99% (*Table 5*).

Experimentally identified epitopes were obtained from the IEDB (*Vita et al., 2015*).

For the multiple Ab test set, 12 Abs were modeled from sequences (*Sela-Culang et al., 2014*) as described above. PDB code 4E9O:X was used for the vaccinia D8 structure. FREAD identified 2AZW:A 87–91 (<u>S</u>NHRQ) as the best match for the missing loop from 207 to 209 (<u>S</u>NHE<u>G</u>), and this structural fragment was grafted into the model as described above.

## Antigen variant expression

Ag variants comprising complete extracellular and transmembrane B7H6 domains were ordered as gBlocks from Integrated DNA Technologies (IDT) and cloned into a HEK surface expression vector (pPPI4). Expression plasmids were transfected into HEK cells using polyethyleneimine as described

previously (*Choi et al., 2015*). Briefly, HEK-293F cells at a density of $10^6$ cells/mL were transfected with antigen variant plasmids at a concentration of 1.0 mg/L each and cultured for 2 days before conducting cell staining experiments and flow cytometry analysis.

## Cell lines

HEK-293F cell line was purchased from ThermoFisher (Catalog #R79007). Cell lines were not verified or tested for mycoplasma contamination after purchase.

## Fluorescent staining for flow cytometry

Fluorescent staining of cells was performed as described previously (*Choi et al., 2015*). Briefly, 96-well plates containing $2.5 \times 10^5$ cells/well of HEK cells expressing designed Ag variants were washed 3x with PBS + 0.1% BSA (PBS-F) before a primary incubation for 1 hr with 100 nM TZ47, PB11 scFv-Fc, NKp30-Ig, H48 (negative control Ms Ab), or PG9 (negative control Hu Ab). Cells were then washed 3x with PBS-F before incubation with either Anti-Mouse-AlexaFluor488 or Anti-Hu-Alexa-Fluor647 secondary Abs for 20 min. After a final wash with PBSF, cells were re-suspended in 200 μL PBSF +Propidium Iodide (PI) to stain for dead cells. Live cell gates were drawn based on FSC vs. SSC and negative PI staining and only this population was used for determination of AlexaFluor-488 or −647 signal. These cells were then separate into binding (fluorescence) positive or negative populations, as transient transfections generally include a population of HEK cells that did not uptake any plasmid. Relative integrated mean fluorescence intensity (I-MFI) values of binding positive cells were calculated as the product of (% of total cells) x (geometric mean fluorescence intensity). To normalize for varying expression levels introduced by differences in transfection efficiency, the normalized relative I-MFI for each Ab was calculated by dividing by the relative I-MFI of NKp30-Ig binding to each design. To normalize for WT binding, normalized relative I-MFIs for each variant were divided by the normalized relative I-MFI of WT B7H6.

## Data availability

The EpiSweep software used as the basis for implementing EpiScope is available under an academic-use license and may be accessed at http://www.cs.dartmouth.edu/~cbk/episweep. The general method and detailed instructions for installing and using EpiSweep have been previously published (*Choi et al., 2017*; *Parker et al., 2013*). Additional inputs for EpiScope, enabling reproduction of the design results reported here, are provided in the supplementary, source data, and source code files that accompany the article.

---

# Additional information

## Competing interests

Chris Bailey-Kellogg: Dartmouth faculty and a co-member of Stealth Biologics, LLC, a Delaware biotechnology company. This author acknowledges that there is a potential financial conflict of interest related to his associations with this company, and he hereby affirms that the data presented in this paper is free of any bias. This work has been reviewed and approved as specified in Chris Bailey-Kellogg's Dartmouth conflict of interest management plans. The other authors declare that no competing interests exist.

## Funding

| Funder | Grant reference number | Author |
| --- | --- | --- |
| National Institutes of Health | R01 GM098977 | Chris Bailey-Kellogg |
| National Research Foundation of Korea | 2016H1D3A1938246 | Yoonjoo Choi |
| National Science Foundation | CNS-1205521 | Chris Bailey-Kellogg |
| National Institutes of Health | 5F30 AI122970-02 | Casey K Hua |
| National Institutes of Health | 1R01AI102691 | Margaret E Ackerman |

| Center of Biomedical Research Excellence | 8P30GM103415 | Charles L Sentman Margaret E Ackerman |
| Allan U. Munck Education and Research Fund at Dartmouth | | Charles L Sentman Margaret E Ackerman |

The funders had no role in study design, data collection and interpretation, or the decision to submit the work for publication.

### Author contributions
Casey K Hua, Conceptualization, Data curation, Funding acquisition, Investigation, Methodology, Writing—original draft; Albert T Gacerez, Investigation, Writing—review and editing; Charles L Sentman, Conceptualization, Supervision, Funding acquisition, Investigation, Writing—review and editing; Margaret E Ackerman, Chris Bailey-Kellogg, Conceptualization, Supervision, Funding acquisition, Investigation, Methodology, Writing—original draft; Yoonjoo Choi, Conceptualization, Data curation, Software, Funding acquisition, Investigation, Methodology, Writing—original draft

### Author ORCIDs
Casey K Hua [iD] http://orcid.org/0000-0002-4420-7228
Margaret E Ackerman [iD] http://orcid.org/0000-0002-4253-3476
Yoonjoo Choi [iD] http://orcid.org/0000-0001-9687-8093
Chris Bailey-Kellogg [iD] http://orcid.org/0000-0003-1860-0912

### Decision letter and Author response
Decision letter https://doi.org/10.7554/eLife.29023.045
Author response https://doi.org/10.7554/eLife.29023.046

## Additional files
### Supplementary files
• Source code 1. TINKER minimization key file.
DOI: https://doi.org/10.7554/eLife.29023.038

• Source code 2. OSPREY configuration files.
DOI: https://doi.org/10.7554/eLife.29023.039

• Supplementary file 1. PyMol session file for an example of 2VXT as shown in *Figure 1*.
DOI: https://doi.org/10.7554/eLife.29023.040

• Supplementary file 2. PyMol session file for B7H6 binding disruptive designs against TZ47 and PB11.
DOI: https://doi.org/10.7554/eLife.29023.041

• Supplementary file 3. Full sequences of TZ47, PB11, and all B7H6 variants.
DOI: https://doi.org/10.7554/eLife.29023.042

• Transparent reporting form
DOI: https://doi.org/10.7554/eLife.29023.043

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
