## [Decision Letter]

Thank you for submitting your article "Computationally-driven Identification of Antibody Epitopes" for consideration by *eLife*. Your article has been reviewed by three peer reviewers, and the evaluation has been overseen by a Reviewing Editor and Arup Chakraborty as the Senior Editor. The following individual involved in review of your submission has agreed to reveal his identity: Timothy A Whitehead (Reviewer #1).

The reviewers have discussed the reviews with one another and the Reviewing Editor has drafted this decision, integrating all points from the individual reviews, to help you prepare a revised submission. We have also considered the possibility that this work may best be considered in our Tools and Resources (TR) category rather than as a Research Article (RA). That distinction may hinge on your responses as to why this method appears to work as that may contribute a fundamental understanding beyond the obvious value of your method.

Summary:

The manuscript describes a procedure, termed EpiScope, that combines structural modeling and limited experimentation to localize epitopes on an antigen of known, or inferred, 3D structure for one or more antibodies known to bind to said antigen.

The paper contains 5 main parts, most of them outlined in the Results section.

1) Description of EpiScope.

2) Prospective epitope localization for two antibodies against the target B7H6.a) Analysis of individual binders with two independent set of antigen mutant designs.b) Simultaneous analysis of both antibodies.

3) Retrospective analysis of 33 antibody: antigen complexes from the PDB.

4) Epitope binning and epitope localization for 12 antibodies to the D8 envelope protein of vaccinia virus.

5) Comparison of EpiScope results to the PEASE method (Sela-Culang et al., 2014). NB. This portion is presented in the Discussion, but given its nature (new results and data), it would make more sense to present it in Results.

Overall, the manuscript describes what could be a potentially significant step towards solution of an important problem with wide practical and fundamental implications. This method fills a nice niche between fine epitope mapping by comprehensive mutagenesis and deep sequencing, and rough epitope binning experiments conducted by expression and binding characterization of individual antigen domains.

The revision calls for additional detail and disclosure to establish more clearly the scope and limitations of the methodology, when applicable attempt an understanding of why the method seems to work as well as it does, and last, but not least, make method details more transparent as to enable interested independent researchers to reproduce the key aspects of this work. The items below, collected from the three independent reviewers, aim to specify the requests explicitly.

Essential revisions:

1) On the method, which may be described briefly as follows: Ab models are generated using the PIGS webserver, and docking decoys are generated using the rigid body docking software program ClusPro. In a key step, Ag variants are computationally designed to maintain stability and disrupt binding modes of a subset of the clustered decoys. These variants are constructed and experimentally characterized, and – as needed – follow-up experiments are designed to obtain fine epitopes. The last, key, step is not described in enough detail to allow an understanding, let alone set the stage for independent implementation, of the method.

2) The link provided in the transparent reporting file is currently a placeholder (as an aside, the referred to work, Choi et al., 2017, was first published online in late 2016; thus, unclear why the link remains a placeholder at this time); one can hope this would provide the lacking detail in the future, but right now it makes it harder to evaluate the work.

3) Nonetheless, some basic detail on the actual cases (see below) could help fill the gaps and provide real insight into how the method works. The authors should also disclose limitations of the method (more on this below), particularly the impact of errors in antibody modeling, conformational change in antigen upon binding (a factor not taken into account in the rigid-body docking approach of ClusPro), errors in ClusPro and docking procedures in general. Readers of the paper would benefit by a frank discussion of how far one can push this methodology in terms of accuracy of models for antigen: for example, choose Ag-Ab sets where Ag comparative models give average TM scores <0.82 (as many models do), and discuss the limitations in pure modeling approaches in determining accuracy of their method in retrospective examination; presumably it may be hard to assess whether mutations will disrupt Ag stability, when using a model for the Ag, a point the authors do not address.

4) In more detail: for example, docking of each of the two antibodies (TZ47 and PB11) to the antigen B7H6 resulted in 28 docking models, and yet only 4 and 5 triple mutants were needed to determine the epitopes. It is understood that the epitope is determined if a set of mutations disrupts binding. However, does the result mean that 27 of the 28 potential models were eliminated as binding outside the epitope? Or do the 28 docked models define only 4 or 5 distinct interfaces? Based on Figure 2 the latter could be the case. However, according to Figure 3—figure supplement 1, the docked models together define a very large fraction of the antigen surface. How can the 4 or 5 mutant designs cover this entire surface? Are the 3 mutations within each triple mutant far from each other to provide the coverage? If this is the case, how well is the final epitope defined? What fraction of the surface is covered by the docking models, and what surface fraction is the final epitope? These questions should be answered to have a better understanding of how and why the method works. Also, it is not clear how the three mutations in the triple mutants are placed relative to each other. Spelling out the sequences of all designed mutants tested could go a long way towards clarifying these issues (see below).

5) Further methodology questions that need to be addressed. For docking: (a) How are the representative docking models selected (centroid, lowest energy)? (b) Does the number of docking clusters influence the final number of designs? (c) Average computational time required for variant design should be reported, and (d) ClusPro option should be "non-CDR masking." For variant design and clustering: Which implementation of the k-medoids algorithm has been adopted? For mutant design choices, how are "related sequence information" (for the antigen) and energy considerations balanced? In the 1H0D example around Figure 4—figure supplement 3; it is said that "since the loop has no mutational information in closely related protein sequences (panel C), mutations that could disrupt binding are not considered in the design process." This could be a real limitation whenever such "related sequences" information is lacking or restricted for a given antigen of interest; one would think the energy-based criteria to choose mutations could compensate in such cases. This issue deserves some discussion, if anything, as avenue for future research.

6) On the prospective test for two B7H6 antibodies, the PDB code for the antigen is given, but not the sequences of the antibodies. It's possible this information is available for TZ47, as it has been described in the literature previously; this does not seem to be the case at all for the second antibody, PB11. To support the data in Figure 2 and Figure 3, it is essential to spell out the 4 (for TZ47) and 5 (for PB11) B7H6 mutant designs; this could easily be included in a supplemental Excel file, for example. Data for the 4 TZ47-relevant designs are presented in Figure 2, but not what they are in much detail. For PB11, data for only one design (of the 5 apparently made and tested) is shown in Figure 2-F. In Figure 2, binding of the Nkp30 ligand (used to monitor intact folding) appears reduced by about 5-fold relative to the wild type B7H6, and this is not explained or commented on by the authors. By contrast, the designs made to probe binding of TZ47 appear to maintain close to 100% of the WT binding to the control ligand. A key potential benefit of the method for mutant design is the expectation that it will flag positions (and associated mutations) most likely to disrupt binding by antibody, while least likely to affect antigen folding, stability and expression, as to enable a straightforward interpretation of a "loss of or reduced binding" readout. It is thus essential to address what may appear as first blush as a weakness in this aspect.

7) Similar disclosure is needed to support Figure 3: the exact sequences of the "integrated set of 6 (B7H6) designs" are essential part of the work. Having the data for all 6 (in the equivalent of Figure 3) would also be informative and would allow a more transparent evaluation of the whole data set.

8) Last, in Figure 2—figure supplement 1, binding of ligand to WT B7H6 appears bimodal; why is this and how it may impact interpretation of data obtained with the mutants?

9) On the analysis of 33 antibody: antigen complexes, characterization of antibody models should be made more in line with how these are evaluated in the literature (see for example issue 8 of volume 83 of Proteins on antibody modeling assessment). Minimally, the rms. deviations in CDRH3s need to be reported. It would also make the work significantly more "reader-friendly" if details like H3 sequences, and lengths, organism of origin of antibody, and so on would be added to Figure 4—table supplement 1 or related table in the CSV-formatted supplementary material.

10) Upon cursory analysis, it appears that 20 of the 33 antibodies in the set are of mouse origin, 2 from rat and 11 are human (or humanized). As antibodies (and perhaps binders in general using alternative scaffolds) of other species or origins (more unique synthetic designs, for example) become more prevalent, their structures will be harder to model, before the PDB "catches up," and this is a point worth mentioning in the paper.

11) Given that the docking poses tend to cover a large portion of each of the antigen's surface, it is hard to envision how the method can narrow down to a handful of proposed mutants that in close to 90% of the test cases include a residue that is part of the Ab: Ag interface. Working out at least one example in detail, including publication, as supplementary data, of the docked Ab: Ag models as PDB files and so on, could help illustrate how this level of success may be achieved. The fact the method does not suffer significantly when using moderate quality models of the antigen as well suggests some low-resolution feature of molecular recognition, as encoded in the respective amino acid sequences of the interacting partners, is in play here. This aspect requires more discussion.

12) On the selection of the 33 cases: using the same criteria adopted by the authors (70% max sequence identity for the antigen; 70% maximum sequence identity for the antibody; max resolution 3.0; protein antigens) in the generation of the retrospective test sets (with the exception the sequence length), one would have retrieved 76 complexes from the SabDab database. This is a larger list and about 2/3 of the structures in the test set of 33 are absent from the new query. Is this a function of database growth since the work was done or do some other filtering criteria need to be specified?

13) On epitope binning and localization of 12 antibodies to the D8 antigen. This section applies EpiScope to a case study from Sela-Culang et al., 2014. At first blush, some of the results are striking, but under closer examination of the sequences of the 12 antibodies (which, to make the paper more reader-friendly, should be disclosed in supplementary material), it is apparent some of the "predictions" of same epitope are rather non-surprising (see also Jia et al. JIM 2004, 288: 91-98; PMID: 15183088, for some background on this notion). Should anyone be surprised, for example, that antibodies AB12.2 and CC7.1 compete with one another, given they have identical CDR H3s and very similar sequences for the other CDRs? Similarly, for FH4.1 and LA5. Some of this result is hinted at in Figure 5—figure supplement 2 (and associated data), but it demands some effort from the reader to sort out. One should also note that the H3s of EB2.1 and JE10 have identical lengths and differ in just one position, while the H3s of BH7.2 and JA11.2 differ by just 2 – other CDRs in those pairs present high similarity as well. Not surprisingly, again, each of these pairs map to the same epitope group. Lastly, the structure of LA5 in complex with the D8 antigen was solved (PDB code 4EBQ), which may be interesting to comment on. The reality here is that of the 12 mAbs, only a fraction may be considered truly independent, after very simple examination of CDR3 sequences (HC and LC), and it may make sense to redo the analysis using an independent subset. The statement commented on below (subsection “Computationally-driven epitope binning and localization of multiple Abs targeting the same Ag”, end of third paragraph) could be modified or illuminated with this consideration.

14) On the comparison of EpiScope and PEASE for the different test cases. First, to reiterate, we suggest including this section in Results instead of Discussion. Second, and this could be in the Discussion, the authors should elaborate on the potential signal present in the "lower resolution" PEASE approach and on the statement "[…] suggesting a potential benefit to incorporating PEASE predictions into the generation of hypotheses for EpiScope-directed experimental validation."

---

## [Author Response]

Summary:The manuscript describes a procedure, termed EpiScope, that combines structural modeling and limited experimentation to localize epitopes on an antigen of known, or inferred, 3D structure for one or more antibodies known to bind to said antigen.The paper contains 5 main parts, most of them outlined in the Results section.1) Description of EpiScope.2) Prospective epitope localization for two antibodies against the target B7H6.a) Analysis of individual binders with two independent set of antigen mutant designs.b) Simultaneous analysis of both antibodies.3) Retrospective analysis of 33 antibody: antigen complexes from the PDB.4) Epitope binning and epitope localization for 12 antibodies to the D8 envelope protein of vaccinia virus.5) Comparison of EpiScope results to the PEASE method (Sela-Culang et al., 2014). NB. This portion is presented in the Discussion, but given its nature (new results and data), it would make more sense to present it in Results.

As suggested, we have moved the comparison to the Results section.

Overall, the manuscript describes what could be a potentially significant step towards solution of an important problem with wide practical and fundamental implications. This method fills a nice niche between fine epitope mapping by comprehensive mutagenesis and deep sequencing, and rough epitope binning experiments conducted by expression and binding characterization of individual antigen domains.

We thank the reviewers for the nice characterization of where and how we are contributing to the study of antibody:antigen recognition. We have incorporated into the Discussion the phrase describing the niche filled.

The revision calls for additional detail and disclosure to establish more clearly the scope and limitations of the methodology, when applicable attempt an understanding of why the method seems to work as well as it does, and last, but not least, make method details more transparent as to enable interested independent researchers to reproduce the key aspects of this work. The items below, collected from the three independent reviewers, aim to specify the requests explicitly.

We agree that it is very interesting that, in general, computational modeling and protein design seem to work as well as they do in leveraging key determinants of recognition in order to efficiently localize epitopes. We augmented our analysis of both the prospective and retrospective tests, as detailed below, in order to provide additional insights into when and why this is possible. We likewise certainly want to make the method transparent and widely useable, so as to enable a wide range of other such investigations, and have addressed the suggestions for how to do so.

Essential revisions:1) On the method, which may be described briefly as follows: Ab models are generated using the PIGS webserver, and docking decoys are generated using the rigid body docking software program ClusPro. In a key step, Ag variants are computationally designed to maintain stability and disrupt binding modes of a subset of the clustered decoys. These variants are constructed and experimentally characterized, and – as needed – follow-up experiments are designed to obtain fine epitopes. The last, key, step is not described in enough detail to allow an understanding, let alone set the stage for independent implementation, of the method.

This is a nice overall description of the method. We agree that the design of follow-up experiments is not described in any detail; that is because this step is in fact not the focus of the work here, and would not contribute to testing the overarching premise that computational modeling and design can leverage information about Ab-Ag binding encoded in sequence in order to drive experiments that efficiently identify epitope residues. Thus, while additional, finer-grain information could be obtained by follow-up experiments, we focus here on the initial, coarse-grained characterization as a novel contribution to the study of Ab-Ag recognition, as was nicely described by the reviewers in the summary. We have augmented the Discussion (“resolution” subsection) with some further thoughts about how future investigations may pursue this topic.

2) The link provided in the transparent reporting file is currently a placeholder (as an aside, the referred to work, Choi et al., 2017, was first published online in late 2016; thus, unclear why the link remains a placeholder at this time); one can hope this would provide the lacking detail in the future, but right now it makes it harder to evaluate the work.

We have been responding to requests for that software by email rather than automatically by web form, since the demand for deimmunized proteins, the main use case for the platform, is not (yet) sufficiently high to be bothersome. We have updated the page accordingly.

3) Nonetheless, some basic detail on the actual cases (see below) could help fill the gaps and provide real insight into how the method works. The authors should also disclose limitations of the method (more on this below), particularly the impact of errors in antibody modeling, conformational change in antigen upon binding (a factor not taken into account in the rigid-body docking approach of ClusPro), errors in ClusPro and docking procedures in general. Readers of the paper would benefit by a frank discussion of how far one can push this methodology in terms of accuracy of models for antigen: for example, choose Ag-Ab sets where Ag comparative models give average TM scores <0.82 (as many models do), and discuss the limitations in pure modeling approaches in determining accuracy of their method in retrospective examination; presumably it may be hard to assess whether mutations will disrupt Ag stability, when using a model for the Ag, a point the authors do not address.

The retrospective tests illustrated some of these limitations, and we have further elaborated the discussion of our observations there, as well as more generally in a new “limitations” subsection in the Discussion.

a)We agree with the reviewers’ concern that errors in Ab modeling could detrimentally affect the results. We were not able to see that with these retrospective tests since, despite using turn-key methods, the models were all quite good (we have added quantification to the text). Consequently there were no discernable trends. In fact, one of the most accurately modeled Ab structures, 1FE8 (CDR-H3 RMSD: 0.63A) failed due to lower quality docking (fnat: 0.1, just above ‘acceptable’ docking quality), while one of the worst models (2XQB: CDR-H3 RMSD 7.21A) had higher quality docking and succeeded (fnat: 0.32, above ‘medium’ quality docking). We have elaborated this in the Discussion (“Limitations | Ab homology models”), and with the new Figure 6—figure supplement 1.

b) Likewise, we agree that if there is major conformational change in binding, the docking and thus epitope localization could be led astray. And more generally, an incorrect Ag model (cf. point d) could negatively impact how much signal can be extracted. It depends overall on how well the actual binding interfaces are accurately modeled (among other factors). We have added suitable discussion (“Limitations | Ag homology models”).

c) While successes and limitations of docking are well studied in the field in general, we have elucidated this point in the context of this study. In summary, we observe that docking quality is affected by model quality (crystal structures are better), and the epitope localization failure cases all had poor docks (as assessed by f_nat_). We have added Figure 6 and discussion (“Limitations | Ab:Ag docking models”).

d) The degree of freedom under our control here, and indeed observable in prospective situations, is the sequence identity between the query and the template. I.e., rather than attempting to somehow force the average TM-score to be worse, we more realistically chose templates of moderate sequence identity. When dealing with a model in practice, one wouldn’t know TM-score and would have to proceed (or not) on the basis of sequence identity. Thus, we feel that this test set gives a reasonable characterization of what one could expect in practice, and we have further justified this approach in the methods. We found that in this study, there was no significant observable impact on results from Ag modeling quality. As one specific example, epitopes of one of the most poorly modeled targets (4JZJ) were still identifiable. We have added discussion (“Limitations | Ag homology models”) and included Table 5 with the sequence identities to further help elucidate this point.

e) The reviewers are correct that there is not sufficient data for these test cases to assess impacts of mutations on antigen stability. And indeed that is not our contribution here – we use one of many well-established computational techniques for predicting and designing for stability; this one in particular has been successfully applied to other challenging cases such as mutation of hydrophobic core residues in our previous publications. Note that furthermore, our objective here is not to improve stability, but to ensure that the target not significantly destabilized while making putative binding-disruptive mutations. Perhaps mutagenesis for epitope identification is one of the best-case scenarios for such cases: the mutations are fairly far apart, at solvent-exposed positions, and using amino acids that appear in the sequence record. Thus the predicted effects are good, and even intuitively we would expect them to be benign (cf. Alanine scanning). We have added related discussion (“Limitations | Ag mutations”). Finally, we note that for the prospective tests, the experimental results show that the designed mutations did not destabilize the antigen to the point of either eliminating expression or binding of its natural ligand.

4) In more detail: for example, docking of each of the two antibodies (TZ47 and PB11) to the antigen B7H6 resulted in 28 docking models, and yet only 4 and 5 triple mutants were needed to determine the epitopes. It is understood that the epitope is determined if a set of mutations disrupts binding. However, does the result mean that 27 of the 28 potential models were eliminated as binding outside the epitope? Or do the 28 docked models define only 4 or 5 distinct interfaces? Based on Figure 2 the latter could be the case. However, according to Figure 3—figure supplement 1, the docked models together define a very large fraction of the antigen surface. How can the 4 or 5 mutant designs cover this entire surface? Are the 3 mutations within each triple mutant far from each other to provide the coverage? If this is the case, how well is the final epitope defined? What fraction of the surface is covered by the docking models, and what surface fraction is the final epitope? These questions should be answered to have a better understanding of how and why the method works. Also, it is not clear how the three mutations in the triple mutants are placed relative to each other. Spelling out the sequences of all designed mutants tested could go a long way towards clarifying these issues (see below).

These are a good set of related questions, and we have updated the text to make the method and the interpretation of the results clearer in that regard. We note that the main goal is to identify the general area of the Ag that is interacting with the Ab (in that mutations disrupt binding) – the docking models are a means to the end, not the end. So we do not try to eliminate all models, and in fact, there are multiple models consistent with the disruptive mutations. This is true even with a single identified mutation, which in our retrospective set (Figure 4—figure supplement 4; see below) would effect on average ~12 docking models (with footprints covering ~47% of the surface).

As the reviewer suggests, the mutations are not immediately adjacent to each other; in the main results as reported in the initial submission, they were limited to be on average (across the set of mutations in a design) 12Å apart or less. However, this distance varied in the different cases according to the selection algorithm. We elaborated the text and associated tables (Table 1 and Table 2) and files (Supplementary file 2 – sequences, and Supplementary file 3 – pymol session) for the prospective application to characterize the patterns of mutations, docking models, and footprints.

We further leveraged the retrospective tests to systematically explore impacts of distances and numbers of mutations on these characteristics, using just a single design (set of mutations) to enable direct assessment of the impact of the parameters, adding Figure 4—figure supplement 4 and related text (“Generalizability | inter-mutation distances”). First (panel (A)), we examined the actual distribution of inter-mutation distances in designs (under the constraint on their maximum), and found it to peak at 11~15Å for both two and three mutation designs. We then studied (panels B-D) the impact of varying the distance threshold, both on the success of hitting an epitope and on the “resolution” in terms of number of consistent docking models and number of residues in their footprints.

The success rate of the single mutation designs was approximately 25%. Using two or three mutations helps, achieving a 50-55% success rate for a single double or triple mutant. Spreading mutations further out than our initial 12 Å limit didn’t seem to help, and in fact could hurt (perhaps due to lack of localization). Bringing them too close together likewise decreased the success rate, though we note that there are few designs available at shorter cut-offs (particularly 6 Å). Lastly, the distance cut-off affects resolution in the way that might be expected – a larger cut-off yielded a larger “footprint”, both in terms of consistent docking models and in terms of their footprints on Ag surfaces. The trends were fairly linear. As noted above, even a single mutation was consistent with about 12 docking models hitting about 47% of the surface. The double and triple mutants hit a few more docking models and thus a bit more of the surface, but didn’t sacrifice too much above that baseline for the large increase in success rate mentioned above. While the cut-off does result in a trade-off between the resolution and the success rate, the threshold we used in the initial results (12Å) gave one of the best success rates at all mutational loads we have tested.

5) Further methodology questions that need to be addressed. For docking: (a) How are the representative docking models selected (centroid, lowest energy)? (b) Does the number of docking clusters influence the final number of designs? (c) Average computational time required for variant design should be reported, and (d) ClusPro option should be "non-CDR masking." For variant design and clustering: Which implementation of the k-medoids algorithm has been adopted? For mutant design choices, how are "related sequence information" (for the antigen) and energy considerations balanced? In the 1H0D example around Figure 4—figure supplement 3; it is said that "since the loop has no mutational information in closely related protein sequences (panel C), mutations that could disrupt binding are not considered in the design process." This could be a real limitation whenever such "related sequences" information is lacking or restricted for a given antigen of interest; one would think the energy-based criteria to choose mutations could compensate in such cases. This issue deserves some discussion, if anything, as avenue for future research.

We have further elaborated these details. The reviewers’ text included letters (a) and (b); we continued the pattern in order to match question and response.

a) Selection of representative docking models was handled by the ClusPro webserver for these results. By default, ClusPro generates ~70,000 models which are then clustered into ~30 representatives (in antibody mode). We used all the docking models provided by ClusPro, rather than selecting just the centroid or just the lowest energy one, and treated them all as equally plausible. We have clarified this matter in the text (“Methods | EpiScope”).

b) While the number of docking models of many targets was close to 30, there was enough variation to allow evaluation of the correlation between the number of docking models and the number of final selected designs. For crystal structures, the correlation coefficient (Kendall’s τ) was 0.397, and 0.443 for model structures (both p-values < 0.01). We have added to the Discussion (“Limitations | Ab: Ag docking models”) specifics from this experience along with more general thoughts about the relationship between the number of models and computational effort.

c) While the timing depends on many factors related to other discussion points (number of docking models, number of allowed mutations, etc.), and the parts specific to EpiScope are simply written as Python scripts and not tuned for speed, we note that the test targets all required only about a day (using 10 nodes on a cluster) for the key steps of generating and clustering the designs. We have added this to the Discussion (“Limitations | computational cost”) as part of the general characterization of trade-offs and limitations.

d) Right, we have fixed the terminology regarding our use of the “non-CDR masking” option.

e) The K-medoids implementation uses a single file from the pyclust package, which is based on the partitioning around medoids algorithm. We modified the script to deal with the Hausdorff distance. We have clarified this point in the text.

f) As is often done to limit the degrees of freedom in protein design and focus on those mostly likely to work (as nature has already evaluated them), the mutations considered are filtered a priori to be those appearing among homologs. There is subsequently no explicit balance between sequence statistics and energies, as it is assumed that accepted mutations are all okay if they are energetically favorable. We have elaborated this point in the text (“Limitations | Ag mutations”).

g) We agree that the selection of allowed mutations from homologs could be relaxed or even dropped in practice, e.g., by evaluating individual energies as suggested. The example the reviewer points to illustrates the limitation of adopting the filter, and we have added a brief discussion as suggested (“Limitations | Ag mutations”).

6) On the prospective test for two B7H6 antibodies, the PDB code for the antigen is given, but not the sequences of the antibodies. It's possible this information is available for TZ47, as it has been described in the literature previously; this does not seem to be the case at all for the second antibody, PB11. To support the data in Figure 2 and Figure 3, it is essential to spell out the 4 (for TZ47) and 5 (for PB11) B7H6 mutant designs; this could easily be included in a supplemental Excel file, for example. Data for the 4 TZ47-relevant designs are presented in Figure 2, but not what they are in much detail. For PB11, data for only one design (of the 5 apparently made and tested) is shown in Figure 2-F. In Figure 2, binding of the Nkp30 ligand (used to monitor intact folding) appears reduced by about 5-fold relative to the wild type B7H6, and this is not explained or commented on by the authors. By contrast, the designs made to probe binding of TZ47 appear to maintain close to 100% of the WT binding to the control ligand. A key potential benefit of the method for mutant design is the expectation that it will flag positions (and associated mutations) most likely to disrupt binding by antibody, while least likely to affect antigen folding, stability and expression, as to enable a straightforward interpretation of a "loss of or reduced binding" readout. It is thus essential to address what may appear as first blush as a weakness in this aspect.

a) Uploaded in Supplementary file 3.

b) Mutations for each design were added as Table 1. We have also included the mutations for PB-specific designs in this table. Multi-Ab designs are included in Table 2.

c) One EpiScope design was tested for PB11 – by the time we had added PB11 as a test case for EpiScope, we had actually tested PB11 binding to all TZ47-specific designs and Macaque-Human B7H6 Chimeras and had roughly localized its epitope to the N-terminal region of Domain I. Thus based on the four PB-specific designs generated by EpiScope, we cloned and tested the one design with mutations in that region.

d) The reviewers are correct in pointing out the impact on NKp30 binding of the disruptive PB11-Ag2 design as compared to the TZ47 designs, which may relate to the positions of the mutations within each design relative to the binding footprint of NKp30. The NKp30 binding region is very close to the mutations contained in the PB11-Ag2 design at the N-terminal region of Domain I (Figure 2). Thus, although the mutations contained in the PB11-Ag2 design do not fall within the NKp30 binding footprint, it is possible that they had a regional effect in decreasing NKp30 binding. In contrast, the TZ47-specific designs were further away from the NKp30 binding region (Figure 2). We recognize the confusion this causes in using NKp30 binding as a marker of stability/expression, and have clarified this in the manuscript.

7) Similar disclosure is needed to support Figure 3: the exact sequences of the "integrated set of 6 (B7H6) designs" are essential part of the work. Having the data for all 6 (in the equivalent of Figure 3) would also be informative and would allow a more transparent evaluation of the whole data set.

The sequences for the integrated designs are uploaded in Supplementary file 3. The mutations are summarized in Table 2. Once again, only designs that contained mutations in epitope localized regions (based on binding data to other EpiScope designs and macaque-human chimeras) were created and tested.

8) Last, in Figure 2—figure supplement 1, binding of ligand to WT B7H6 appears bimodal; why is this and how it may impact interpretation of data obtained with the mutants?

We are unsure as to the origin of this bimodal behavior, though it has been reported/observed previously. If we were to speculate, the RMA-B7H6 cells used included both adherent and suspension cells, which may differ in their clustering of surface proteins including B7H6. Because NKp30 is more reliant upon avidity than TZ47 and PB11, it is possible that the bimodal distribution results from either the adherent or suspension population having greater clusters of B7H6 on their surface, enabling greater NKp30-binding.

9) On the analysis of 33 antibody: antigen complexes, characterization of antibody models should be made more in line with how these are evaluated in the literature (see for example issue 8 of volume 83 of Proteins on antibody modeling assessment). Minimally, the rms. deviations in CDRH3s need to be reported. It would also make the work significantly more "reader-friendly" if details like H3 sequences, and lengths, organism of origin of antibody, and so on would be added to Figure 4—table supplement 1 or related table in the CSV-formatted supplementary material.

These model details are now provided in Table 4; implementation details are now in the Materials and methods. Note that the IMGT definition was used for CDRs, and RMSDs were evaluated using all backbone atoms (N, C, Cα and O). Accuracy ranges were similar to those reported in [Choi and Deane, Molecular Biosystems 7.12 (2011): 3327-3334] -- sub-Å for non CDR-H3 loops and ~2Å for CDR-H3.

10) Upon cursory analysis, it appears that 20 of the 33 antibodies in the set are of mouse origin, 2 from rat and 11 are human (or humanized). As antibodies (and perhaps binders in general using alternative scaffolds) of other species or origins (more unique synthetic designs, for example) become more prevalent, their structures will be harder to model, before the PDB "catches up," and this is a point worth mentioning in the paper.

We have now incorporated further discussion of the general state of antibody modeling (“Limitations | Ab homology models”), including extension to those from other origins and alternative antibody-like protein binders that may be harder (or indeed may be easier).

11) Given that the docking poses tend to cover a large portion of each of the antigen's surface, it is hard to envision how the method can narrow down to a handful of proposed mutants that in close to 90% of the test cases include a residue that is part of the Ab: Ag interface. Working out at least one example in detail, including publication, as supplementary data, of the docked Ab: Ag models as PDB files and so on, could help illustrate how this level of success may be achieved. The fact the method does not suffer significantly when using moderate quality models of the antigen as well suggests some low-resolution feature of molecular recognition, as encoded in the respective amino acid sequences of the interacting partners, is in play here. This aspect requires more discussion.

We discussed the overarching question of resolution in the response to point 4, above. We completely agree that the study here reveals that modeling and design can indeed extract and leverage “low-resolution features of molecular recognition, as encoded in the respective amino acid sequences of the interacting partners” (another nice phrase we thank the reviewer for and have incorporated into the Discussion). To illustrate in detail, we have provided in Supplementary file 1 pymol session file that includes all 13 docking models and associated designs for the example in Figure 1. We also have included Supplementary file 2 for all docking models and designs for the B7H6 prospective test.

12) On the selection of the 33 cases: using the same criteria adopted by the authors (70% max sequence identity for the antigen; 70% maximum sequence identity for the antibody; max resolution 3.0; protein antigens) in the generation of the retrospective test sets (with the exception the sequence length), one would have retrieved 76 complexes from the SabDab database. This is a larger list and about 2/3 of the structures in the test set of 33 are absent from the new query. Is this a function of database growth since the work was done or do some other filtering criteria need to be specified?

We initially had 70 complexes, so indeed the database seems to have grown by 6. There were a few additional quality filtering criteria that were accidentally dropped from the text – we require a single Ag chain with no missing backbone atoms, so that homology modeling would not impact the results. Furthermore, we limited Ag length to be between 50 and 300 residues. We have specified these criteria in the revised manuscript (“Methods | Retrospective test sets”).

13) On epitope binning and localization of 12 antibodies to the D8 antigen. This section applies EpiScope to a case study from Sela-Culang et al., 2014. At first blush, some of the results are striking, but under closer examination of the sequences of the 12 antibodies (which, to make the paper more reader-friendly, should be disclosed in supplementary material), it is apparent some of the "predictions" of same epitope are rather non-surprising (see also Jia et al. JIM 2004, 288: 91-98; PMID: 15183088, for some background on this notion). Should anyone be surprised, for example, that antibodies AB12.2 and CC7.1 compete with one another, given they have identical CDR H3s and very similar sequences for the other CDRs? Similarly, for FH4.1 and LA5. Some of this result is hinted at in Figure 5—figure supplement 2 (and associated data), but it demands some effort from the reader to sort out. One should also note that the H3s of EB2.1 and JE10 have identical lengths and differ in just one position, while the H3s of BH7.2 and JA11.2 differ by just 2 – other CDRs in those pairs present high similarity as well. Not surprisingly, again, each of these pairs map to the same epitope group. Lastly, the structure of LA5 in complex with the D8 antigen was solved (PDB code 4EBQ), which may be interesting to comment on. The reality here is that of the 12 mAbs, only a fraction may be considered truly independent, after very simple examination of CDR3 sequences (HC and LC), and it may make sense to redo the analysis using an independent subset. The statement commented on below (subsection “Computationally-driven epitope binning and localization of multiple Abs targeting the same Ag”, end of third paragraph) could be modified or illuminated with this consideration.

This dataset is the one Sela-Culang and colleagues used in their demonstration of the new paradigm of antibody-specific epitope prediction, and in fact they integrated the binning data into epitope prediction by combining predicted antigenic “patches” for the antibodies within a bin, thereby exploiting this seemingly redundant information from highly similar antibody sequences to improve predictions. So we followed their lead in looking both at the relationships within and between bins, and were pleased that the Episcope “binning” results already largely reflected the observed ones, and might be complementary in additional ways. It is true that CDR sequences drive CDR structures which drive Ab-Ag binding, and thus binding similarity can often be largely explained by CDR sequence similarity. However, it is of course not always obvious how strong an inference about binding similarity can be made from sequence similarity or even overall structural similarity. For example, while it appears that here heavy chain CDRs largely drive the binding (by comparison to binning patterns, as suggested by the reviewer), there are certainly cases where light chain ones do [Ko et al., PloS one 10.7 (2015): e0134600]. And likewise, there are different degrees of similarity among the different CDR-Hs. As a result it is not clear how to combine them to infer impacts, or to identify which CDRs matter and why, or to define what the impacts of mutations on binding. These are all explicitly modeled and dealt with in a consistent fashion in our approach. We have added results and discussion to that effect (“[…] multiple Abs targeting the same Ag”).

Following the suggestion for more independence among representative antibodies, we reanalyzed the data using a subset of 7 of the 12 antibodies with no identical CDR sequences, and replaced Figure 5—figure supplement 3 with the new analysis both with EpiScope and with sequence analysis alone. (We note that Figure 5 was also updated to have the correct labels on the x axis of the matrix, which were in reverse order before.) While CDR-H sequence similarities still largely explain the binning similarity, EpiScope does appear to pull out additional information. EpiScope integrates structural modeling of the Ab, along with Ab-Ag docking and Ag design to disrupt putative binding, thereby systematically integrating all the relevant information and proposing hypotheses for experimental evaluation. We have further elaborated this analysis in the text (“[…] multiple Abs targeting the same Ag”).

We have added Figure 5—figure supplement 2 to the manuscript as the reviewer suggested. Please note that 4EBQ is not a complex structure but 4ETQ is. There are five designs generated for D8-LA5 using EpiScope and two of them are in the Ab-Ag binding interface.

14) On the comparison of EpiScope and PEASE for the different test cases. First, to reiterate, we suggest including this section in Results instead of Discussion. Second, and this could be in the Discussion, the authors should elaborate on the potential signal present in the "lower resolution" PEASE approach and on the statement "[…] suggesting a potential benefit to incorporating PEASE predictions into the generation of hypotheses for EpiScope-directed experimental validation."

We have moved the comparison to the Results.

We have also briefly discussed (end of that section) how PEASE predictions may complement EpiScope, by further focusing docking and design efforts on those regions likely to be fruitful, either as determined by purely in silico modeling, or by integrating additional prior data.